# Automated NMR resonance assignments and structure determination using a minimal set of 4D spectra

Thomas Evangelidis[1], Santrupti Nerli[2,3], Jiří Nováček[1], Andrew E. Brereton[4], P. Andrew Karplus[4], Rochelle R. Dotas[5], Vincenzo Venditti[5,6], Nikolaos G. Sgourakis[3] & Konstantinos Tripsianes[1]

Automated methods for NMR structure determination of proteins are continuously becoming more robust. However, current methods addressing larger, more complex targets rely on analyzing 6–10 complementary spectra, suggesting the need for alternative approaches. Here, we describe 4D-CHAINS/autoNOE-Rosetta, a complete pipeline for NOE-driven structure determination of medium- to larger-sized proteins. The 4D-CHAINS algorithm analyzes two 4D spectra recorded using a single, fully protonated protein sample in an iterative ansatz where common NOEs between different spin systems supplement conventional through-bond connectivities to establish assignments of sidechain and backbone resonances at high levels of completeness and with a minimum error rate. The 4D-CHAINS assignments are then used to guide automated assignment of long-range NOEs and structure refinement in autoNOE-Rosetta. Our results on four targets ranging in size from 15.5 to 27.3 kDa illustrate that the structures of proteins can be determined accurately and in an unsupervised manner in a matter of days.

[1] CEITEC—Central European Institute of Technology, Masaryk University, Kamenice 5, Brno 62500, Czech Republic. [2] Department of Chemistry and Biochemistry, University of California Santa Cruz, Santa Cruz, CA 95064, USA. [3] Department of Computer Science, University of California Santa Cruz, Santa Cruz, CA 95064, USA. [4] Department of Biochemistry and Biophysics, 2011 Ag & Life Sciences Bldg, Oregon State University, Corvallis, OR 97331, USA. [5] Department of Chemistry, Iowa State University, 2438 Pammel Drive, Ames, IA 50011, USA. [6] Roy J. Carver Department of Biochemistry, Biophysics and Molecular Biology, Iowa State University, Ames, IA 50011, USA. Correspondence and requests for materials should be addressed to N.G.S. (email: nsgourak@ucsc.edu) or to K.T. (email: kostas.tripsianes@ceitec.muni.cz)

Nuclear magnetic resonance (NMR) structure determination relies on recording a network of nuclear Overhauser enhancement (NOE) restraints from multidimensional spectra[1]. Obtaining near-unambiguous assignments of long-range NOEs is challenging due to substantial overlap in the spectra which becomes more pronounced for larger proteins. This is typically addressed through first establishing the chemical shift assignments of backbone and sidechain atoms using multiple (6–10) triple-resonance spectra[2, 3], which are then used as anchors to guide the assignment of NOEs during iterative structure refinement[4]. State-of-the-art tools such as FLYA[5], PINE[6] and UNIO[7] can automate the resonance assignment and structure determination process. In principle, recording a smaller number of higher dimensionality spectra can provide a complementary approach to increase signal dispersion and resolve ambiguities[8]. With the emergence of non-uniform sampling and reconstruction methods, such datasets can be recorded in reasonable time[9]. Recent approaches for automated resonance assignments based on three- and four-dimensional (3D and 4D) NOE data make use of a known structure to guide the assignment process[10, 11]. However, for de novo structure determination, further development is needed to perform resonance assignments at the high levels of completeness and correctness that are required for NOE data-driven structure determination.

Recent methods allow for structure modeling guided by NMR chemical shifts, used as a means to optimize a physically realistic energy function that reproduces the native features of protein structures[12, 13]. Chemical shift Rosetta (CS-Rosetta) relies on backbone assignments along with Rosetta's Monte Carlo fragment assembly protocol to model protein structures in the 10–12 kDa range[12]. CS-Rosetta was superseded by resolution-adapted structural recombination (RASREC)-Rosetta, extending the size limit to 25 kDa using backbone residual dipolar couplings (RDCs) and amide NOEs[14], or to 40 kDa using sparse NOE data acquired on methyl-labeled, perdeuterated NMR samples, assigned manually[15]. In addition, the use of evolutionary information in conjunction with NMR chemical shift data can be used to model protein targets in the 25–40 kDa range[16–18]. Finally, autoNOE-Rosetta performs automated assignment of long-range NOEs and structure refinement using iterations of parallel RASREC-Rosetta calculations[19]. In all these methods, the use of advanced conformational sampling methodologies enables protein structure determination using a sparse network of restraints[20]. However, a significant bottleneck remains in establishing correct sidechain assignments at sufficient completeness levels to drive the automated assignment of long-range NOEs[20]. Moreover, the use of methyl-labeled samples requires extensive deuteration, which can be challenging for several biologically important systems[21].

Here we combine the powerful autoNOE-Rosetta approach with a new automated assignment algorithm (4D-CHAINS) in a complete pipeline for NMR structure determination. First, 4D-CHAINS utilizes two complementary experimental datasets, a 4D-TOCSY (Total Correlated Spectroscopy) and a 4D-NOESY (Nuclear Overhauser Effect Spectroscopy), to obtain near-complete resonance assignments of backbone and sidechain $^1$H, $^{13}$C and $^{15}$N atoms. The resonance lists provided by 4D-CHAINS form the basis for iterative assignment of long-range NOEs and structure determination using autoNOE-Rosetta, which exploits through-space correlations recorded in two 4D-NOESY datasets, one amide to aliphatic, and one aliphatic to aliphatic. The combined approach allows us to obtain structural ensembles for proteins up to 27 kDa, without the need for deuteration or selective labeling, by leveraging the well-resolved spectral features of the 4D datasets together with Rosetta's energy function. Our NMR data and detailed analysis, performed for one benchmark case with known X-ray structure and three additional blind

targets, illustrate that the new approach can consistently deliver high-resolution structural ensembles of biologically relevant proteins by greatly reducing the number of required experiments and human time spent.

## Results

**Development of the 4D-CHAINS assignment algorithm.** Towards developing 4D-CHAINS, we recorded for four different protein targets of size from 15.5 to 27.3 kDa, a 4D HC(CC-TOCSY(CO))NH, and a 4D $^{13}$C,$^{15}$N edited HMQC-NOESY-HSQC (HCNH) experiment. The largest protein target of size 27.3 kDa was chosen based on its apparent correlation time of ~15 ns that still allows for TOCSY transfer to occur (Supplementary Figure 1). We also recorded a 4D $^{13}$C,$^{13}$C edited HMQC-NOESY-HSQC (HCCH) experiment to further assist in structure determination. To address the assignment problem, 4D-CHAINS uses 2D probability density maps of correlated $^{13}$C–$^1$H chemical shifts to effectively identify possible spin systems (Fig. 1, Supplementary Figure 2). In particular, 4D-CHAINS combines sequential information present in the 4D-HCNH TOCSY and intraresidue information present in 4D-HCNH NOESY $^{13}$C–$^1$H planes, respectively, by clustering TOCSY or NOESY peaks to Amino Acid Index Groups (AAIGs) via their common $^{15}$N–$^1$H frequency (Supplementary Figure 3). 4D-CHAINS computes probability scores at several steps (amino acid-type prediction, sequential AAIG relations based on TOCSY–NOESY connectivities, alignment of peptides to the protein sequence) to yield a confidence score for a given AAIG being assigned to a specific protein residue. Finally, 4D-CHAINS uses an Overlap Layout Consensus (OLC) assembly approach adopted from genome assembly[22] to match continuous AAIG segments along the protein sequence (Fig. 2). The final assignment solutions are consistent with both the joined probability score and the OLC model.

A uniform 4D-CHAINS protocol was applied to all four targets (Fig. 3a). The algorithm mapped correctly all AAIGs to the respective protein sequences with >95% completeness (Supplementary Figure 4). The TOCSY-based assignments alone covered approximately 80% of all aliphatic chemical shifts with an error rate of <0.5% (Fig. 3b, Supplementary Figure 5). To increase the overall assignment completeness, we obtained additional information from the HCNH NOESY spectrum by employing the concept of common NOEs between successive residues[21]. The TOCSY–NOESY combination enabled more complete assignments with 94% correct aliphatic chemical shifts and a combined error rate of 1.3% (Fig. 3b, Supplementary Figures 5–7, Supplementary Table 1). The concept of common NOEs in obtaining assignments was further tested by providing 4D-CHAINS fixed $^{15}$N–$^1$H assignments and the HCNH NOESY spectrum alone (Supplementary Figure 8). This assignment scenario (NOESY) allows users to extend existing backbone assignments, obtained conventionally, to cover sidechain atoms with 86% accuracy and an error rate of approximately 5% (Fig. 3b, Supplementary Figure 5).

**Performance of 4D-CHAINS relative to existing methods.** To test the performance of 4D-CHAINS relative to existing assignment programs, we performed calculations using a popular assignment method, FLYA[5], for all protein targets used in the current study. While 4D-CHAINS relies exclusively on the combination of 4D-HCNH TOCSY and 4D-HCNH NOESY, the FLYA algorithm is designed to combine peak patterns from any number of input spectra. Therefore, we provided FLYA with all available spectra (4D-HCNH TOCSY, 4D-HCNH NOESY, 4D-HCCH NOESY). Notwithstanding, 4D-CHAINS outperforms FLYA consistently for all four protein targets in our benchmark.

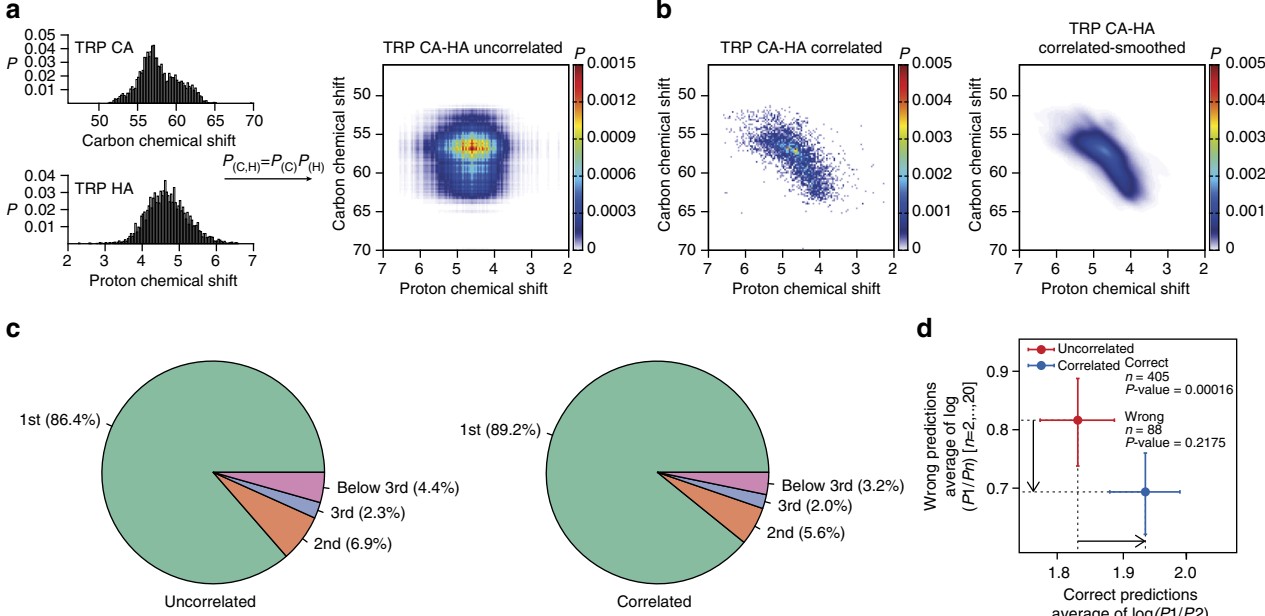

**Fig. 1** Comparison of uncorrelated and correlated chemical shifts probabilities and their power in amino acid-type prediction. **a** The 1D probability distributions for the $C_\alpha$ and $H_\alpha$ atoms of Trp (left) and their joint probability distribution (right). **b** The 2D probability distribution of correlated $C_\alpha$–$H_\alpha$ chemical shifts of Trp (left) and the corresponding smoothed 2D probability density map (right) after applying a Gaussian kernel function. Graphs were generated from the VASCO database with a bin size of 0.04 p.p.m. for protons and 0.2 p.p.m. for carbons. The color gradient scaling differs for (**a**) uncorrelated and (**b**) correlated distributions. **c** Pie charts displaying the ranking of amino acid-type predictions in the current dataset (four proteins) using uncorrelated (left) or correlated distributions (right) of $^{13}$C–$^{1}$H chemical shifts. **d** Analysis of common (correct or wrong) predictions made by using uncorrelated (red) or correlated distributions (blue) of $^{13}$C–$^{1}$H chemical shifts. Horizontal axis plots the mean value and standard error of correct predictions (P1) to the second probable (P2), and vertical axis plots the mean value and standard error of wrong predictions (Pn) to the first probable (P1). Pairwise $t$-test shows that the improvement in predictions made by using correlated instead of uncorrelated $^{13}$C–$^{1}$H chemical shifts (arrows) is statistically significant for the correct ones (P-value = 0.00016) but not for the wrong ones (P-value = 0.2175). The discrepancy is attributed to the relatively small number of wrong predictions available for the test (405 vs 88)

For three proteins, namely RTT, ms6282 and Enzyme I (nEIt), FLYA outputs 90% correct assignments with 7-8% error rate, while for α-lytic protease (aLP) the number of correct assignments is limited to 25% (Supplementary Figure 5). Finally, we manually inspected and extended the 4D-CHAINS results to establish the maximum number of highly accurate assignments for all $^{13}$C–$^{1}$H correlations that can be observed in our 4D spectra (>98%), as a "best-effort" resonance list requiring a modest time investment by a trained user. These supervised assignment lists also contain aromatic and sidechain amide chemical shifts, not considered by the automated 4D-CHAINS protocol (Supplementary Figure 6).

**Iterative structure calculations using autoNOE-Rosetta**. We evaluated the performance of assignments obtained using 4D-CHAINS in driving Rosetta structure determination of a 20 kDa target, aLP, for which several known X-ray structures are available in the Protein Data Bank[23]. Inspection of the X-ray structural ensemble shows a highly complex all-β fold, with two sub-domains each containing a 6-stranded antiparallel β-sheet. In order to establish a "best effort" limit of the Rosetta automated NOE assignment and structure determination protocol, we first performed autoNOE-Rosetta calculations[19] using the supervised assignments together with both NOE datasets (HCNH+HCCH). Additionally, we carried out automated 4D-CHAINS/autoNOE-Rosetta structure calculations (Fig. 3a, Supplementary Figure 9) under four different scenarios, as described above (TOCSY–NOESY or NOESY assignments, each using HCNH alone or HCNH+HCCH NOEs). To evaluate the quality of the

resulting structural ensembles, we used the following criteria: (i) fraction of residues converged within 2.0 Å backbone heavy atom root-mean-square deviation (RMSD) in the final ensemble, (ii) average Rosetta all-atom energies and (iii) RMSD to X-ray structure. Since the Rosetta energy function[24] has a global minimum at the native fold, lower-energy models should also exhibit higher convergence towards the native structure. We observed a good correlation between Rosetta all-atom energy, degree of convergence and structural accuracy (correlation coefficient of 0.93, Fig. 3c). Specifically, using the supervised assignments and HCNH or HCNH+HCCH NOEs, we obtained highly converged structural ensembles (>98%, computed over the core secondary structure regions) that are within 0.7 Å RMSD from the X-ray structure.

Notably, ensembles calculated using the 4D-CHAINS (TOCSY–NOESY or NOESY) automated assignments using both NOE datasets also achieved a high level of convergence (>90%), to within 1.3 and 1.7 Å RMSD from the X-ray, respectively. Using the same 4D-CHAINS assignment lists and the HCNH NOEs alone, the accuracy relative to the X-ray was slightly reduced to 1.7 and 1.9 Å, respectively, while the convergence decreased to approximately 86%, but the models still recapitulated the protein fold and β-sheet topology. This trend is highlighted in a superposition of the lowest-energy aLP model sampled in each calculation on the X-ray structure reference (Fig. 3d).

**Convergence of aLP structures towards the X-ray reference**. To evaluate the relative accuracy and precision of NOE-driven structure determination approaches using different input

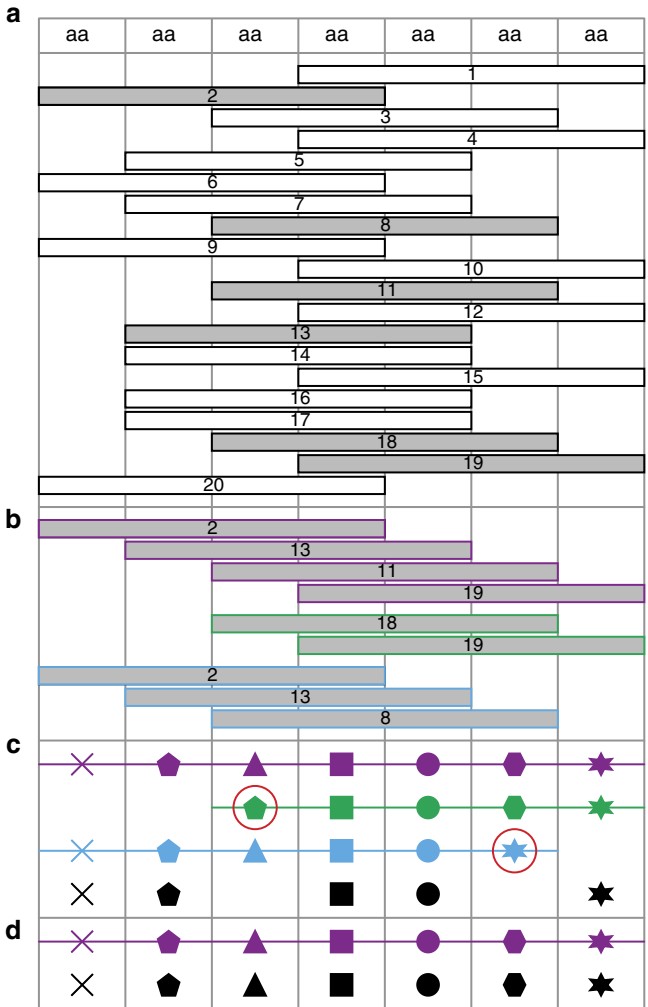

**Fig. 2** Schematic representation of 4D-CHAINS decision making using Overlap Layout Consensus assembly. **a** Chains of AAIGs of given length *L* are translated to peptides and aligned to the query sequence. **b** Contigs of overlapping chains (overlap length *L*-1) are generated. Peptides that do not form contigs are discarded. **c** Contigs are lined up and absolute consensus AAIGs (shapes) are identified. If there is agreement with the associated confidence score (for details see Methods), then AAIGs are assigned to the corresponding amino acids of the sequence (black shapes). **d** Assigned AAIGs are restrained in next rounds of decision making. False peptides and contigs do not longer form and using the same rules additional AAIGs are assigned (triangle, polygon)

assignments and NOE datasets (HCNH or HCNH+HCCH), we performed a single joint Ensemblator[25, 26] analysis of the 12 resulting aLP structural ensembles (six each using Rosetta or CYANA[27]) and a set of 51 X-ray structures from the CoDNaS[28] database (see Methods). A dimensionality-reduced visualization of the relationships between the models (Fig. 4a) reveals that the Rosetta models are consistently closer to the X-ray models than the corresponding CYANA models generated using the same datasets. Overall, the Rosetta models show better convergence, and convergence for all groups correlates strongly with their similarity to the X-ray structures (Fig. 4b). The two Rosetta-generated ensembles based on supervised assignments are nearly equivalent and show the best convergence and greatest similarity to the X-ray structures (Fig. 4a, b). Here, the fragment-based structure refinement in Rosetta allows the generation of highly accurate ensembles from HCNH NOEs alone, which is not

feasible using standard simulated annealing in CYANA. In particular, poor convergence and low similarity to the X-ray ensemble are seen for models calculated by CYANA from the HCNH NOEs alone (Fig. 4a, b; orange and red circles). Conversely, the Rosetta ensembles produced from a single NOESY dataset (HCNH) are in good agreement with the X-ray ensemble (Fig. 4c), and quantitative comparison shows that the structural variability pattern along the protein chain is rather similar, although the NMR ensemble typically has a greater variability than the X-ray ensemble (Fig. 4d). These results suggest that the uncertainty of atom positions in solution correlates with variability associated with different crystal packing environments.

**Consistent blind structure determination of protein targets**. To further test our method in a fully unbiased manner we performed blind structure calculations for three additional protein targets, RTT[29, 30], ms6282 and nEIt of sizes 133, 145 and 248 amino acids (aa), respectively (Table 1). To establish a baseline performance, we carried out CS-Rosetta calculations guided by chemical shifts alone[15], as well as reference CYANA calculations using both input NOE datasets (HCNH+HCCH). With the exception of the smallest target (RTT), the resulting CS-Rosetta models failed to converge (Supplementary Figure 10) and instead sampled conformations with sub-optimal energies (Fig. 5; right column, black). Conformational sampling is drastically improved in autoNOE-Rosetta calculations guided by both supervised or automated 4D-CHAINS assignments, and the resulting structural ensembles are very similar for all targets (Fig. 5; left column). For the largest target, the 27.3 kDa Enzyme I from *Thermoanaerobacter tengcongensis*, NOE contacts provided sufficient constraints to elucidate the structure of the individual domains, but the overall orientation of the two domains was not converged due to the lack of contacts at their interface (domain A, defined by residues 1−143 and domain B, defined by residues 144−248) (Supplementary Figure 11). Here, the use of $^{15}N$–$^{1}H$ residual dipolar couplings allowed us to sample further lower energies, and obtain better convergence by restraining the relative orientation of the two domains (Fig. 5d, Supplementary Figure 11d).

Towards evaluating the effect of different levels of assignment completeness on the performance of autoNOE-Rosetta, we carried out benchmark calculations by randomly removing entries from our "best effort" supervised assignment lists for target aLP and found that autoNOE-Rosetta can identify correct protein fold from as low as 60–70% sidechain assignments. In addition, we performed a detailed comparison of assigned NOE contacts and Rosetta energy distributions, relative to control calculations guided by the supervised assignments. We observe that the use of fully automated assignments results in a small decrease in the total number of NOE contacts identified by Rosetta (approximately 80% for all targets). Furthermore, we obtain similar distributions of assigned NOE contacts among residue pairs in the protein sequence (Fig. 5; middle column). The respective lowest-energy models are built using hundreds of automatically assigned, long-range NOE restraints and exhibit a minimal number of violations (1–4%) involving pairs of atoms that are typically within 1 Å from their estimated upper distance limits (Supplementary Table 2). Given that methyl–methyl NOE contacts play a critical role in defining the hydrophobic core of the protein, we found that ~25% of the total contacts identified by autoNOE-Rosetta are contributed by methyl NOEs for structure calculations using supervised or automated 4D-CHAINS assignments (Supplementary Table 3). Finally, the distributions of energies among the 100 best sampled structures are generally shifted relative to RASREC-Rosetta and show good overlap with their supervised counterparts (Fig. 5; right column).

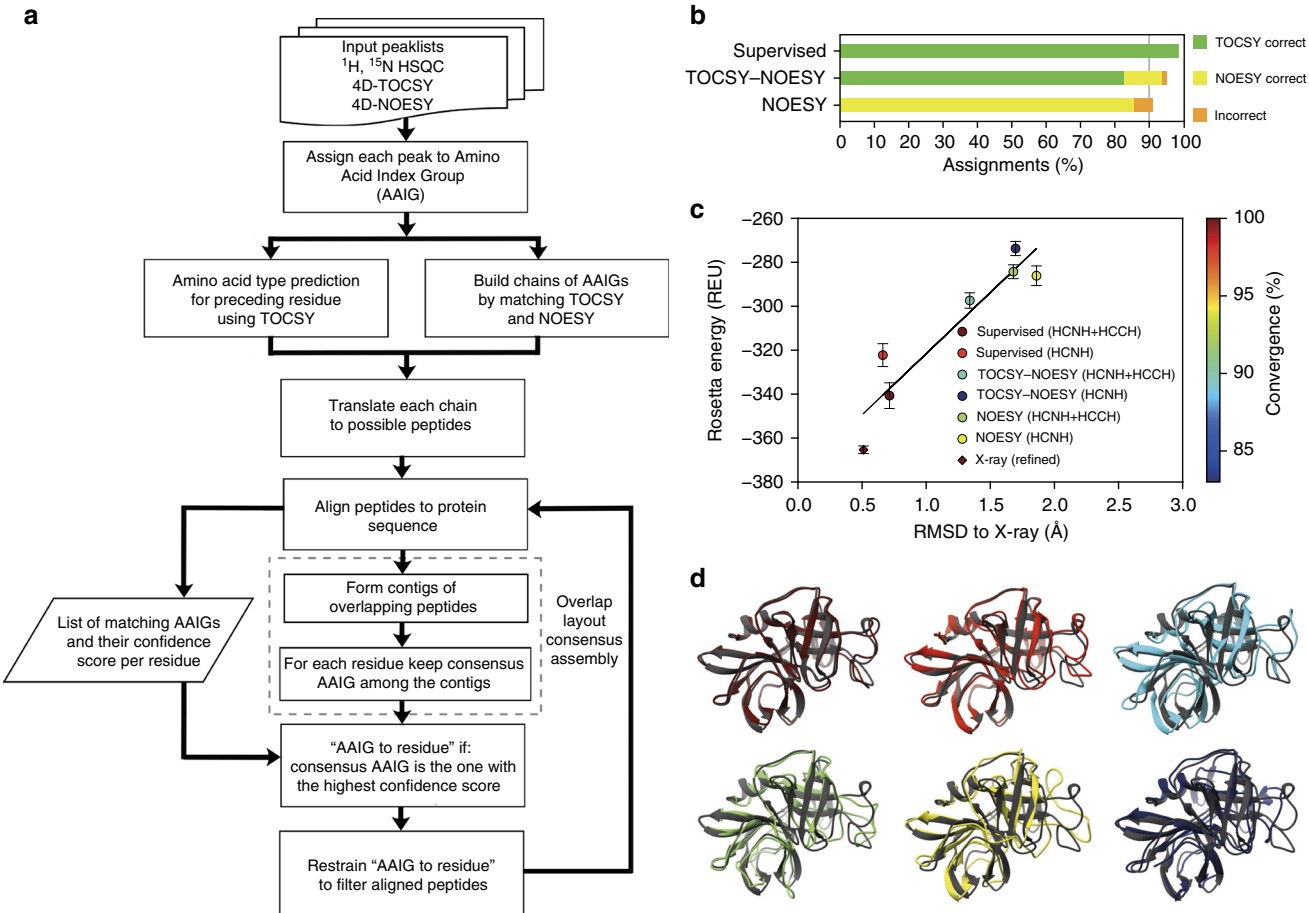

**Fig. 3** Automated structure determination using 4D-CHAINS/autoNOE-Rosetta. **a** Flowchart of the 4D-CHAINS algorithm for automated NMR resonance assignment from two 4D spectra (TOCSY and NOESY). **b** Quality of 4D-CHAINS assignments for supervised, TOCSY–NOESY, and NOESY settings, expressed as the average for the four different protein targets. **c, d** Performance of different 4D-CHAINS assignment scenarios for a 198 aa protein, α-lytic protease, calculated using autoNOE-Rosetta. **c** Goodness of structural ensembles is measured using the Rosetta all-atom energy function, backbone heavy atom RMSD to X-ray structure (PDB ID 1P01) and degree of structural convergence. Average energy values (in Rosetta Energy Units (REU)) for ensembles calculated using indicated data/assignment scenarios, with errors bars shown at 1 standard deviation. Also shown is the average energy of 10 locally refined X-ray structures (diamond)—refinement adapts the X-ray structure to the local optimum of the Rosetta energy, with a minimum change in RMSD (0.5 Å). The color of points represents the average % of converged residues in each ensemble, according to the color scale on the right. **d** Lowest-energy structures in each ensemble (shown in the same color as the points in **c**) superimposed on the X-ray reference structure (gray). Images of structures were produced using Chimera (https://www.cgl.ucsf.edu/chimera)

## Discussion

NMR remains the only biophysical technique that can deliver high-resolution structures of proteins and other biomolecules in their functional, aqueous environment, which constitutes the basis for studying interactions with other molecules and therapeutic compounds. However, standard approaches for NMR resonance assignment rely on recording several complementary datasets which can be limiting for larger, more complex systems due to increased resonance overlap and require a significant time investment by a trained expert to analyze the spectra and establish a complete list of resonance assignments aided by computational tools[31]. Established methods to overcome this problem utilize selective isotopic labeling[32], which can be limiting in terms of the information content present in the NMR data, expensive and challenging to perform for certain systems.

Here, we propose an automated approach for full structure determination using 2–3 4D NMR spectra recorded on a $^{13}$C, $^{15}$N uniformly labeled sample. First, 4D-CHAINS addresses the assignment problem in an efficient and highly robust manner, yielding the correct assignments for at least 95% of residues and

error rates of less than 1.5% (Supplementary Table 1). It is further worth noting that the vast majority of resonances corresponding to sidechain methyls, which are important probes in identifying the protein fold, are correctly assigned by our method. Therefore, the use of 4D-CHAINS allows near-complete assignment of sidechain methyls without the need for site-specific labeling on a perdeuterated background[33] (Supplementary Table 4). Second, autoNOE-Rosetta uses a highly parallelizable iterative algorithm run on a computer cluster to perform assignment of long-range NOEs alongside the structure determination process. The full pipeline takes approximately 10–12 days to execute for a typical protein sample, including the time needed for NMR data acquisition, and requires minimum supervision.

In addition to recapitulating the correct protein fold, autoNOE-Rosetta models obtained using the 4D-CHAINS assignments show accurate placement of sidechains for most residues in the protein structure (Supplementary Table 5). Specifically, close inspection of sidechain conformations in the Rosetta ensembles computed using the supervised assignments shows good overall agreement with the X-ray rotamers for most

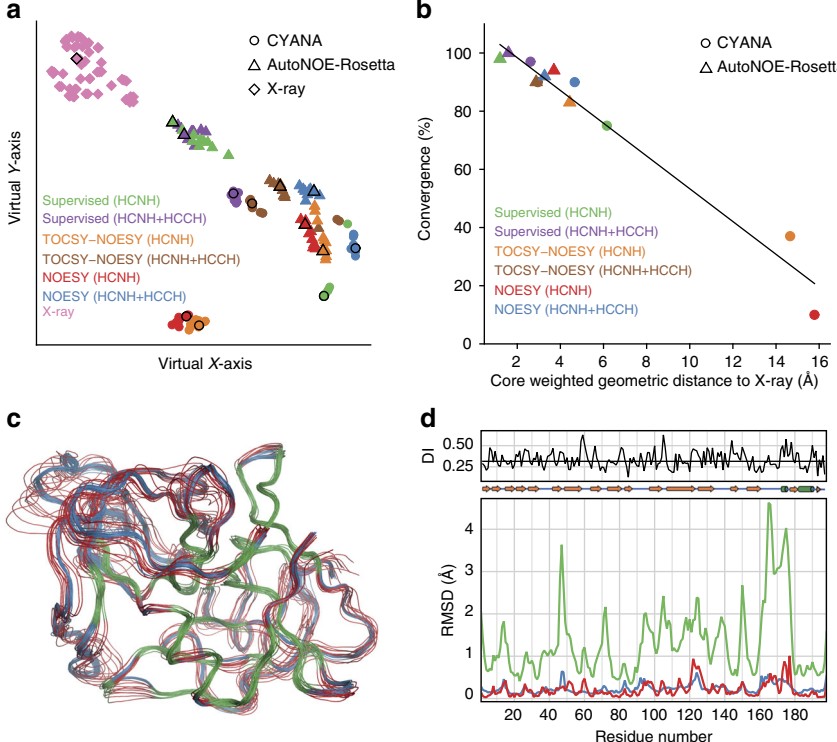

**Fig. 4** Ensemblator[25, 26] analysis of α-lytic protease NMR and X-ray ensembles. **a** t-SNE dimensionality reduction results showing the relationships of the aLP models. The shape and color of each point (see in plot key) convey the method and the data used to generate that model; a black outline highlights the most representative (exemplar) for each type of model. **b** For each type of NMR ensemble, the convergence of the ensemble during refinement is compared to the distance of its exemplar to that of the X-ray ensemble. Shapes and colors of each point (see in plot key) convey the method and the data used to generate that model. The best fit to these data (black line) has an adjusted $R^2$ of 0.96. **c** Wire diagram showing the traces of the backbone heavy atoms for the 4D-CHAINS/autoNOE-Rosetta ensemble obtained using supervised assignments and HCNH NOEs (red) and the X-ray ensemble (blue). The defined core atoms (21.4% of all atoms using a 1.6 Å cutoff) are shown in green. **d** Shown are the per residue global backbone heavy atom variation (bottom panel) for the 4D-CHAINS/autoNOE-Rosetta ensemble obtained using supervised assignments and HCNH NOEs (green) and the X-ray ensemble (blue), along with the closest approach distance for models in each group (red). The discrimination index (top plot, black line) reveals regions of similarity (low values) and difference (high values); the median discrimination (DI) index is indicated by a horizontal line

buried residues (>10 Å² BSA), while using the fully automated assignments results in a small decrease (<10%) in accuracy relative to the models derived using supervised assignments (Supplementary Figure 12 and Supplementary Table 5). Finally, an analysis of long-range NOEs assigned by autoNOE-Rosetta versus predicted from the X-ray structure using a 5.5 Å distance cutoff between all pairs of protons shows good recovery of crystallographic contacts at levels of 67–86%, which are distributed across the entire protein fold (Fig. 6). Taken together, our results underpin that the automated 4D-CHAINS/autoNOE-Rosetta approach yields models that accurately capture the correct global fold as well as atomic features of the native structure.

Overall, the convergence of structures obtained using supervised assignments for the three target proteins, RTT, ms6282 and aLP, are better than or comparable to the convergence of structures obtained using 4D-CHAINS automated assignments, as expected (Table 1). Notably, for Enzyme I, autoNOE-Rosetta achieves a higher level of structural convergence using automated assignments due to enhanced resampling of the correct protein fold during the early stages of the autoNOE-Rosetta structure calculation process. Overall, our results suggest that the fully automated assignment process introduced by 4D-CHAINS has a minimal impact on the performance and quality of the derived structural ensembles by autoNOE-Rosetta, which remain highly consistent with all available input data.

Relative to CYANA, autoNOE-Rosetta can achieve a similar degree of structural convergence using the same input resonance

assignments with both the aliphatic and amide NOE peak lists. Although the total number of structurally degenerate HCNH +HCCH long-range NOE contacts identified by CYANA is higher by (i) ~10% for aLP and ms6282 and (ii) ~25% for RTT and nEIt (Supplementary Figure 13), for three targets, RTT, ms6282 and aLP, the degree of structural convergence achieved by autoNOE-Rosetta is comparable to CYANA; while for the largest target, Enzyme I (nEIt), the autoNOE-Rosetta ensemble is significantly more converged towards the correct fold (Supplementary Figure 14). Generally, the structural ensembles determined using autoNOE-Rosetta are closer to the nearest PDB reference structures by approximately 0.5 Å for RTT, 0.2 Å for ms6282, 0.5 Å for aLP and >2.2 Å for nEIt relative to the structures predicted by CYANA (Supplementary Figure 14). When using the amide NOEs alone together with either automated or supervised resonance assignments, CYANA does not yield converged models, while autoNOE-Rosetta can still deliver models showing the correct protein fold, albeit with reduced convergence relative to calculations performed using both input peak lists, as shown in detail for aLP (Fig. 3) and as outlined for all other targets (Table 1).

In summary, we demonstrate that 4D-CHAINS provides highly accurate and near-complete NMR resonance assignments from two 4D spectra, which are effective in guiding high-resolution structure determination using autoNOE-Rosetta. Our results on four targets in the 15.5–27.3 kDa range indicate that the use of our automated pipeline has a minimal impact on the

**Table 1 Statistics of autoNOE-Rosetta structural ensembles computed using supervised and automated 4D-CHAINS assignments**

| Protein | Assignment data | Peaks | RDCs used | No. of residues | Average Rosetta energy (REU) | [a]Fraction of residues converged (%) | Mean RMSD (Å) to average structure |
|---|---|---|---|---|---|---|---|
| RTT | Supervised | HCNH | | 133 | −239.66 | 98 | 0.86 |
| | | HCNH +HCCH | | | −247.31 | 98 | 0.71 |
| | TOCSY-NOESY | HCNH | | 133 | −243.63 | 99 | 1 |
| | | HCNH +HCCH | | | −243.32 | 95 | 0.94 |
| ms6282 | Supervised | HCNH | | 145 | −263.83 | 92 | 1.15 |
| | | HCNH +HCCH | | | −252.8 | 97 | 0.9 |
| | TOCSY-NOESY | HCNH | | 145 | −262.95 | 93 | 1.19 |
| | | HCNH +HCCH | | | −262.57 | 93 | 0.94 |
| aLP | Supervised | HCNH | | 198 | −324.46 | 99 | 0.84 |
| | | HCNH +HCCH | | | −328.5 | 100 | 0.64 |
| | TOCSY-NOESY | HCNH | | 198 | −274.79 | 86 | 1.49 |
| | | HCNH +HCCH | | | −300.55 | 91 | 1.27 |
| nEIt | Supervised | HCNH | Yes[b] | 248 | −487.79 | 60 | 3.75 |
| | | HCNH +HCCH | Yes[b] | | −475.61 | 84 | 1.4 |
| | TOCSY-NOESY | HCNH | Yes[b] | 248 | −479.54 | 60 | 4.07 |
| | | HCNH +HCCH | Yes[b] | | −491.05 | 91 | 1.36 |

[a] Convergence statistics calculated over core residues
[b] One RDC dataset was used to improve structure convergence
Average Rosetta energies (in Rosetta Energy Units (REU)) reported over 10 lowest-energy structures
All backbone heavy-atom RMSD relative to the average structure calculated over core residues

precision and quality of the resulting structural ensembles, while allowing for a tremendous reduction in human effort and NMR spectrometer time. Lastly, our structural evaluation criteria, in terms of convergence and Rosetta all-atom energy, can clearly distinguish the correct structures, allowing our protocol to be used extensively for generating high-quality models in a truly unsupervised manner. Therefore, our combined approach could be of great practical utility in both high-throughput structural determination projects[34] and NMR-based screening for small-molecule and protein–protein interactions[35].

## Methods

**NMR sample details.** For each uniformly [13]C-, [15]N-labeled protein sample, the concentration, buffer composition and NMR data collection temperature are as follows:

The 0.8 mM RTT in 35 mM potassium phosphate (pH 6.8), 100 mM KCl, 5% $D_2O$, 25 °C.

The 1.2 mM ms6282 in 50 mM sodium phosphate (pH 6.5), 150 mM NaCl, 7% $D_2O$, 25 °C.

The 2.0 mM aLP in 10 mM deuterated sodium acetate (pH 4.0), 50 mM NaCl, 8% $D_2O$, 25 °C.

The 1.8 mM nEIt in 20 mM sodium phosphate (pH 6.5), 100 mM NaCl, 5% $D_2O$, 37 °C.

**NMR data collection.** For each protein target, a set of three sparsely sampled 4D NMR experiments was acquired on 850 or 950 MHz Bruker Avance III spectrometers equipped with $^1H/^{13}C/^{15}N$ TCI cryogenic probehead with z-axis gradients. All NMR spectra were recorded at CEITEC Josef Dadok National NMR Centre using pulse sequences adopted and modified from Bruker library. The 4D HC(CC-TOCSY(CO))NH experiment was acquired with chemical shift evolution performed in semi-constant time manner in $t_1$ ($^1H_{ali}$) and $t_3$ ($^{15}N$) and using FLOPSY16 spin-lock of 12 ms that yielded the best overall signal-to-noise ratio. The spectral widths were set to 12,500 (acq) × 2000 ($^{15}N$) × 8000 ($^{13}C_{ali}$) × 6250 ($^1H_{ali}$) Hz and maximal acquisition times in the indirectly detected dimensions were set to 50 ms for $^{15}N$, 10 ms for $^{13}C_{ali}$ and 16 ms for $^1H_{ali}$. The experiment was acquired with 16 scans per increment and single-scan recycling delay of 1.0 s. The overall number of 1536 points was collected in the acquisition dimension and 2500

hypercomplex points were sparsely distributed over the indirectly detected dimensions. Prior to recording full 4D HC(CC-TOCSY(CO))NH experiment, we recorded the $^{15}N/^1H$ 2D plane of the experiment using a full (incremental) sampling list since our methodology is applicable if the number of signals observed in the 2D plane are ≥50% of expected, based on a standard, sensitivity-enhanced 2D $^{15}N/^1H$ HSQC experiment. In the 4D $^{13}C,^{15}N$ edited HMQC-NOESY-HSQC (HCNH) experiment, the HMQC building block was used to transfer the magnetization between $^1H$ ($t_1$) and $^{13}C$ ($t_2$) with evolution of the $^1H$ chemical shift in semi-constant time manner during both transfer and refocusing of magnetization. The magnetization transfer between $^1H$ ($t_4$) and $^{15}N$ ($t_3$) was designed using a reverse HSQC building block after the 70 ms NOESY mixing time. The data were collected with spectral widths set to 12,500 (acq) × 2000 ($^{15}N$) × 8000 ($^{13}C$) × 10,000 ($^1H$) Hz, respectively. The maximal evolution times in the indirectly detected dimensions were set to 50 ms for $^{15}N$, 10 ms for $^{13}C$ and 20 ms for $^1H$. The experiment was acquired using 1.0 s single-scan recycle delay and 8-step phase cycle with 8 scans per increment. In all, 1536 complex points were acquired in the direct dimension and the overall number of 5000 hypercomplex points was non-uniformly distributed over the indirectly detected dimensions. The 4D $^{13}C,^{13}C$ edited HMQC-NOESY-HSQC (HCCH) experiment uses the same HMQC-type building block as described for the HCNH noesy experiment (see above) within the first $^1H$ ($t_1$), $^{13}C$ ($t_2$) transfer of magnetization. The second $^1H$ ($t_4$), $^{13}C$ ($t_3$) transfer of magnetization following the 70 ms NOESY mixing time is performed using HSQC building block utilizing gradients. Data were collected with spectral widths set to 12,500 (acq) × 8000 ($^{13}C$) × 8000 ($^{13}C$) × 10,000 ($^1H$) Hz, and the maximal acquisition times in the indirectly detected dimensions were set to 20 ms for $^1H$ ($t_1$) and 10 ms for $^{13}C$ ($t_2$, $t_3$). The experiment was acquired with 8 repetitions per increment and 1.0 s single-scan recycling delay. The overall number of 1536 complex points was collected in the acquisition dimension and 5000 hypercomplex points were distributed over the indirectly detected dimensions. For each spectrum the NMR acquisition time was 4 days. From our setup, we can observe that the experimental time needed to acquire three 4D non-uniform sampling spectra is comparable to the total acquisition time of several conventional 3D experiments. However, the analysis of 3D experiments is laborious and further complicated by resonance overlap, which becomes more pronounced with increasing target size. Thus, from the user's standpoint, it is preferable to operate using a pair of complementary experiments which yield the same information in a higher-dimensionality dataset. Finally, our benchmark data illustrate that any additional relaxation losses during the extra chemical shift evolution step needed to acquire the fourth indirect dimension are not prohibitive for highly concentrated samples of stable proteins, which can still yield very rich datasets. All the pulse sequences used in our experiments are available upon request.

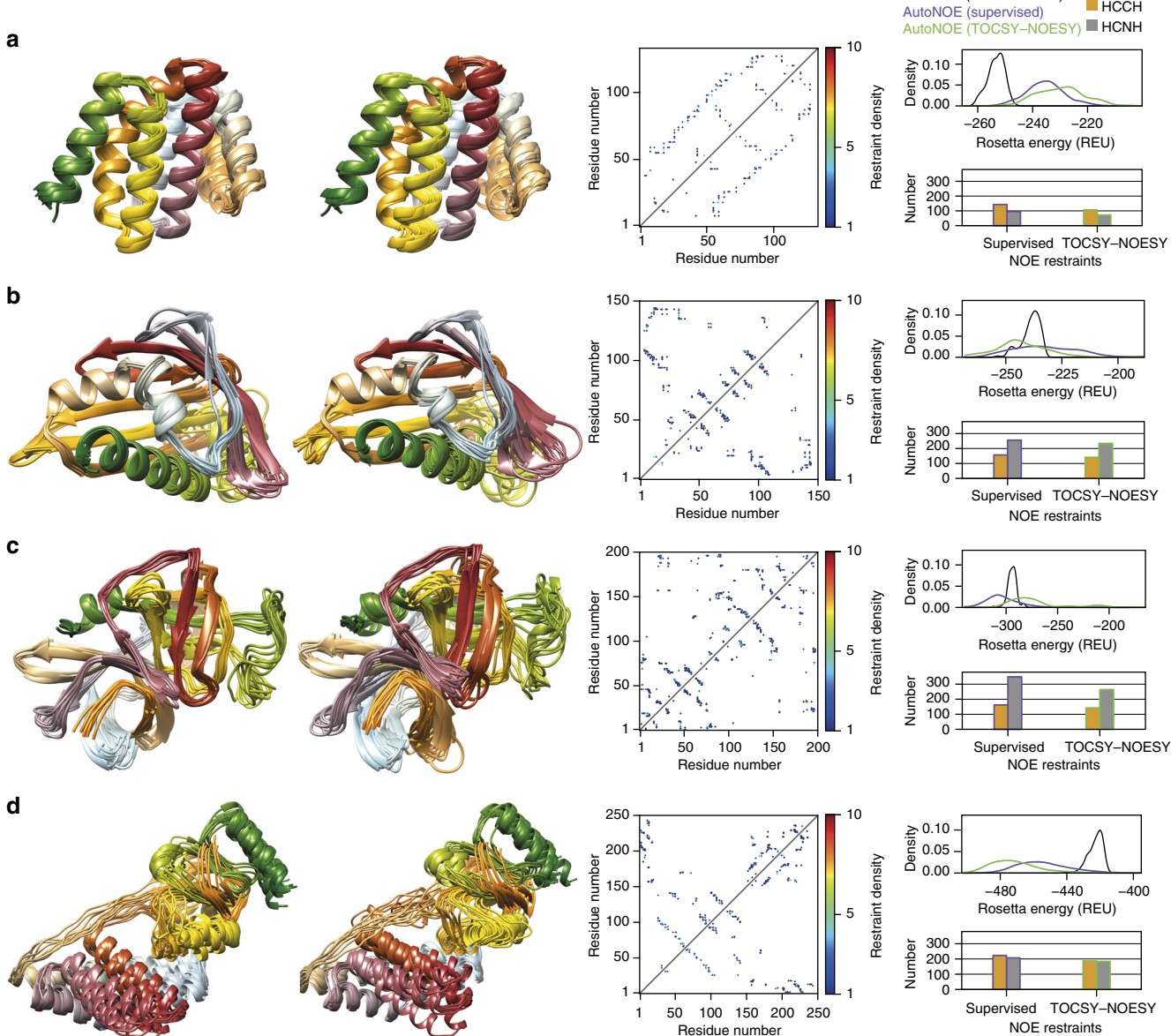

**Fig. 5** Comparison of structural ensembles calculated from supervised versus fully automated assignments. **a** Rtt103 (RTT, 133 aa), **b** KanY (ms6282, 145 aa), **c** α-lytic protease (aLP, 198 aa) and **d** Enzyme I (nEIt, 248 aa). Columns 1 and 2: autoNOE-Rosetta ensembles of 10 lowest-energy structures guided by "best effort" supervised assignments or by automated 4D-CHAINS assignments (TOCSY–NOESY), respectively. Column 3: Sequence map of distance restraints assigned by autoNOE-Rosetta in iterative structure refinement calculations. Here, the upper triangular region shows restraints obtained using supervised assignments, while the lower triangular region using automated 4D-CHAINS assignments. Column 4: Rosetta energy (in Rosetta Energy Units (REU)) distributions and total numbers of assigned long-range restraints. The energy distribution was computed from the 100 lowest-energy structures sampled during the final stage of autoNOE-Rosetta calculations using supervised assignments (purple), 4D-CHAINS assignments (green) and chemical shift fragment-based RASREC-Rosetta calculations without NOEs (black). The bars represent the total number of HCNH (amide to aliphatic) and HCCH (aliphatic to aliphatic) long-range NOE restraints assigned by autoNOE-Rosetta, including ambiguous restraints derived for different stereo-specific groups. RDCs were used to obtain converged Enzyme I structures with respect to the orientation of the two domains reported in row **d**. Images of structural ensembles were produced using Chimera (https://www.cgl.ucsf.edu/chimera)

**Sparse sampling and data processing.** The on-grid Poisson disc sampling[36] was utilized in the present application to distribute individual acquisition points in the indirectly detected dimensions. This sampling scheme introduces distances between the generated time points and has been shown to reduce the level of sampling artifacts in the direct vicinity of signal after the reconstruction[36].

The 4D data were processed using sparse Fourier transform algorithm[37] to check the data quality. Final processing was performed in an iterative manner using the Signal separation analysis approach as implemented in the program cleaner4d[37] (SSA package). Prior to processing with the cleaner4d program, the data were square cosine weighted in the directly acquired dimension and zero-filled to 2 k points using NMRPipe/NMRDraw 3.0[38]. The 4D spectra were analyzed in Sparky[39].

**Peak picking.** Peaks were picked automatically and curated manually using a restricted peak picking strategy. First, the 4D-HCNH NOESY spectrum was picked at a user-defined noise level using both $^{15}$N,$^1$H- and $^{13}$C,$^1$H-HSQC peaks as filters. Then the 4D-HCNH TOCSY spectrum was picked using the 4D-NOESY peaks as filters. Accordingly, all planes were inspected simultaneously in all spectra and picked artifacts were removed. Synchronization of all four shared dimensions in the spectra allows for a highly efficient peak picking and curation process.

**Measurement of RDC restraints for nEIt.** Backbone amide $^1D_{NH}$ RDCs were measured by taking the difference in $^1J_{NH}$ scalar couplings in aligned and isotropic media[40]. The alignment media employed was phage pf1 (16 mg ml$^{-1}$; ASLA

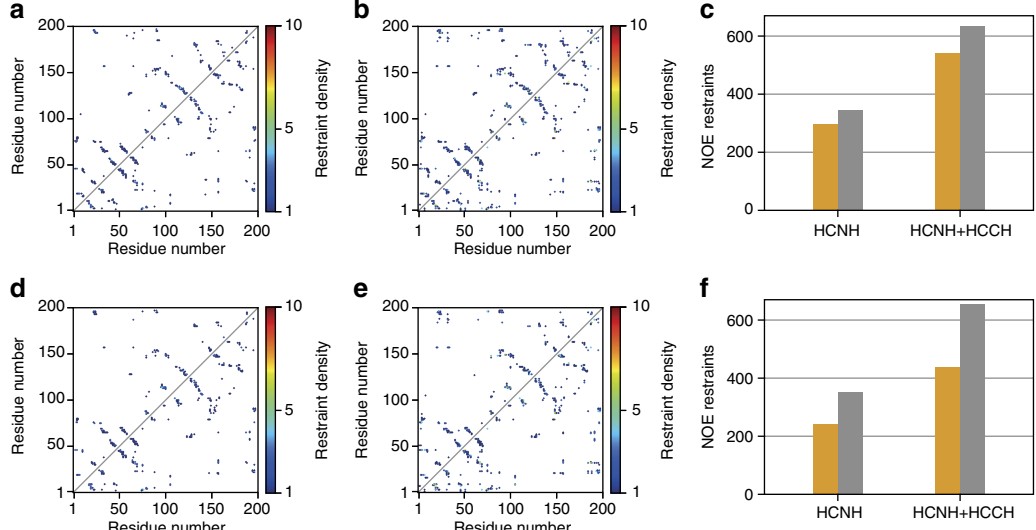

**Fig. 6** Comparison of assigned NOE contacts versus predicted from X-ray structure. Contacts shown as a function of residue pairs along the sequence of α-lytic protease. Upper triangular region shows NOE contacts identified during iterative structure refinement by autoNOE-Rosetta, while that of lower triangular region represents expected contacts as predicted from the X-ray structure (PDB ID 1P01) using a 5.5 Å distance cutoff between all possible proton atom pairs and further removing redundancies due to chemically equivalent protons. Different combinations of input assignments and NOE datasets used are shown as follows. **a** Supervised assignments with HCNH NOEs. **b** Supervised assignments with HCNH+HCCH NOEs. **c** Total number of NOE restraints assigned in **a** and **b** (orange) versus predicted from X-ray structure (gray). **d** 4D-CHAINS TOCSY-NOESY automated assignments with HCNH NOEs. **e** 4D-CHAINS TOCSY–NOESY automated assignments with HCNH+HCCH NOEs. **f** Total number of NOE restraints assigned in **d** and **e** (orange) versus predicted from X-ray structure (gray)

Biotech)[41], and ${}^1J_{NH}$ couplings were measured using the ARTSY pulse scheme[42]. NMR measurements were performed on a Bruker 800 MHz spectrometer equipped with a z-shielded gradient triple-resonance cryoprobe. Spectra were processed using NMRPipe[38] and analyzed using the program Sparky[39].

**Automated resonance assignment using 4D-CHAINS.** 4D-CHAINS is an automated resonance assignment algorithm for backbone and sidechain chemical shifts of proteins. As input it requires the protein sequence in fasta format and peak lists from ${}^1H,{}^{15}N$ HSQC (root), 4D HC(CC-TOCSY(CO))NH and 4D ${}^{13}C,{}^{15}N$ edited HMQC-NOESY-HSQC experiments in sparky format. 4D-CHAINS algorithm tackles the assignment problem in a conventional way[43, 44]. The distinct feature of 4D-CHAINS is that all available aliphatic ${}^{13}C–{}^1H$ coupled frequencies are used for amino acid-type prediction and sequential connectivities, drastically decreasing the ambiguity level (Supplementary Figure 9).

4D-CHAINS overall assignment accuracy depends on chemical shift statistics, currently available in the form of one-dimensional (1D) distributions of proton or carbon resonances for every atom of the 20 amino acids (Fig. 1a). Since the 4D spectra provide direct information on ${}^{13}C–{}^1H$ correlated chemical shifts, we reasoned that statistical correlated chemical shift distributions would improve 4D-CHAINS performance in addressing the assignment problem. For the different ${}^{13}C–{}^1H$ moieties of every amino acid, correlated chemical shifts maps were generated from the VASCO-corrected data[45]. VASCO dataset was chosen instead of the larger BMRB dataset, because chemical shift values of aliphatic carbons that were improperly referenced have been corrected, thus avoiding distortion of the information content used. The resulting 2D probability distributions have bins with zero frequency, due to the relatively small sample size (Fig. 1b). Therefore, we created probability density maps by applying a Gaussian kernel function, given by Eq. 1, to estimate the density at any point

$$G_{(H_0,C_0)} = \frac{1}{2\pi n h_H h_C} \sum_{i=1}^{n} e^{-\frac{1}{2}\left[\frac{(H_i-H_0)^2}{h_H^2} + \frac{(C_i-C_0)^2}{h_C^2}\right]} \qquad (1)$$

where $n$ is the data size, $h_H$ and $h_C$ the bandwidth for the proton and the carbon dimension, respectively. For optimal bandwidth selection we used Scott's rule of thumb $h = n^{-1/6}$. Based on our analysis, the 2D probability density maps of correlated chemical shifts provide improved predictive power when compared to joint probabilities derived from 1D histograms of proton and carbon chemical shifts (Fig. 1c, d).

4D-CHAINS is written in Python programming language and consists of two modules: NH-mapping module and atom-type assignment module. As output, it provides TOCSY and NOESY (intraresidue and sequential) assignments of the input 4D peak lists allowing visual verification of results, and a chemical shift list in XEASY format that can be input together with NOESY peak lists to automated structure determination software.

At first, 4D-CHAINS clusters the 4D-HCNH TOCSY and 4D-HCNH NOESY peaks via the common root resonance (${}^{15}N–{}^1H$) they share to generate AAIGs of ${}^{13}C–{}^1H$ correlated chemical shifts. For a given root resonance, the TOCSY AAIG provides sequential information, that is, the ${}^{13}C–{}^1H$ aliphatic resonances of the previous amino acid in the sequence (i-1), whereas the NOESY AAIG reports on any ${}^{13}C–{}^1H$ moiety that is in close spatial proximity. By virtue of NOE distance dependence, the NOESY AAIG contains most, if not all, of the intraresidue ${}^{13}C–{}^1H$ resonances (i).

For each TOCSY AAIG, 4D-CHAINS calculates the probability of an amino acid type for the preceding residue in the protein sequence using a probabilistic model[46]. Let us denote any amino acid of the 20 types by AA and the set of correlated chemical shifts in a TOCSY AAIG by CCS. The conditional probability $P$(AA|CCS) to get an amino acid type given the observed C–H resonances is highlighted in Eq. 2

$$P(AA|CCS) = \frac{P(CCS|AA)P(AA)}{P(CCS)} \qquad (2)$$

where $P$(CCS|AA) is the conditional probability of ${}^{13}C–{}^1H$ resonances for a given amino acid type, $P$(AA) is the prior probability of finding the given amino acid type in the protein sequence independent of the observed ${}^{13}C–{}^1H$ resonances, and $P$(CCS) is the sum of the $P$(CCS|AA) terms over the 20 amino acid types. $P$(CCS|AA) can be accurately estimated for any amino acid type using the probability density maps of ${}^{13}C–{}^1H$ correlated chemical shifts (Supplementary Figure 2). For a given number of ${}^{13}C–{}^1H$ frequencies in a TOCSY AAIG all permutations of atom-type combinations are considered in calculating the probability for amino acids with possible atom types equal to or larger to the TOCSY frequencies. In practice, however, only a small number of combinations is computed, because many ${}^{13}C–{}^1H$ frequencies have non-zero probability only for distinct atom types of any amino acid (Supplementary Figure 2). $P$(CCS|AA) is considered the most probable combination, expressed as the product of probabilities of each ${}^{13}C–{}^1H$ pair belonging to different atom types of a given amino acid. If the number of TOCSY ${}^{13}C–{}^1H$ pairs is larger than the expected atom types of a given amino acid then $P$(CCS|AA) is set to zero. For every TOCSY AAIG several amino acid-type predictions are made and ranked according to their conditional probabilities. Depending on the TOCSY transfer efficiency, amino acids with long sidechains are predicted rather unambiguously. In our datasets, accurate predictions defined as the correct amino acid type being the most probable reached 89% (Fig. 1c).

Next, sequential connectivity information is obtained by matching the ${}^{13}C–{}^1H$ frequencies of every TOCSY AAIG (i-1) to ${}^{13}C–{}^1H$ frequencies present in any other NOESY AAIG (i), excluding the NOESY AAIG with the same root resonance (${}^{15}N–{}^1H$) as the TOCSY AAIG (Supplementary Figure 3a). The sequential connectivities established for each TOCSY AAIG may vary in occupancy rate, defined as the ratio of matched frequencies versus the total number of TOCSY frequencies (Supplementary Figure 3a). The algorithm creates a directed rooted

tree from each AAIG and adds progressively edges and nodes using the connectivity information, until it reaches a maximum chain length (Supplementary Figure 3b). As a tradeoff between efficiency and memory consumption, maximum length is set to six. Each chain $X$ is then assigned a probability of occurrence given by the product of the probabilities of each connectivity type in that chain as shown in Eq. 3 to estimate the significance of each chain

$$P(X) = \prod_{k=1}^{L-1} P(X_{k \to k+1}) = \prod_{k=1}^{L-1} P(\text{occupancy rate}) \qquad (3)$$

where $L$ is the chain length and $k$ the position in the chain. In principle, chains with higher occupancy rate of connectivities are more likely to be correct.

Subsequently, the chains are used to generate a larger number of peptide sequence segments using the amino acid-type predictions obtained earlier. Each peptide is aligned to the protein sequence using the Needleman–Wunsch algorithm[47]. Many peptides are discarded at this stage due to alignment mismatches. For the aligned peptides, an alignment score $S_{align}$ is computed using the BLOSUM90 similarity matrix[48], which quantifies the importance of the alignment to a specific amino acid sequence. Taken all the above into account, the weighted probability of assigning an AAIG from chain $X$ to a specific residue in the protein sequence is given by Eq. 4.

$$S(X) = P(\text{AA}|\text{CCS}) * P(X) * S_{align} \qquad (4)$$

Since several different chains can be mapped at overlapping positions in the protein sequence, multiple AAIGs may correspond to each protein residue. For each position in the protein sequence, a confidence score ($C_s$) is computed for every AAIG corresponding to the given position by summing over all the chains as indicated in Eq. 5.

$$C_s = \sum_{}^{\text{all chains } X} S(X) \qquad (5)$$

In order to identify the correct AAIGs mapped to the protein sequence from the large pool of aligned chains, the 4D-CHAINS algorithm exploits the overlap information by performing OLC assembly similar to DNA assembly techniques[22] (Fig. 2). From N- to C-terminus of the protein sequence, series of aligned chains are merged to contigs with identical overlap of length $L-1$, where $L$ is the length of the chains. A contig terminates if there is no overlap to extend or if it encounters an AAIG that is already part of it. Chains that cannot be merged to contigs are considered spurious and discarded. Finally, all contigs are aligned to the protein sequence. For an AAIG to be assigned to a given residue in the protein sequence, two conditions must be met. First, only the absolute consensus AAIGs among the different contigs are taken into account for a given position in the sequence and, second, the consensus AAIG must have the higher confidence score for the given position (Fig. 2).

Mapping of AAIGs to the protein sequence is accomplished by a succession of iterations that differ in three parameters used: (i) the length of chains built; (ii) a Z-score cutoff that controls the amino acid-type predictions to be considered per TOCSY AAIG when chains are translated to peptides; and (iii) the occupancy rate of connectivities between a TOCSY AAIG and all matched NOESY AAIGs.

In the first iteration, stringent criteria are applied to ensure greater fidelity of predictions and extract long chains ($L = 6$) that are less likely to be aligned in a wrong position of the sequence. OLC assembly provides an additional level of scrutiny and removes lonely chains that cannot be extended to either end and are likely false. Accordingly, only consensus AAIGs are selected and if there is agreement with the probabilistic model (confidence score), then are mapped to certain positions of the protein sequence. Mapped AAIGs are restrained in successive rounds by eliminating all amino acid-type predictions and connectivities they participate in that are inconsistent with the NH mapping. This reduces noise and allows us to proceed gradually with shorter chains (minimum length 3) to fill short regions in the sequence that are flanked by gaps in connectivities or proline residues, incorporate AAIGs in the sequence that fulfill the connectivity criteria but have low amino acid-type probability due to abnormal chemical shifts, and finally account for the fact that NOESY AAIGs may not match all frequencies of a TOCSY AAIG. In each round both the OLC and the probabilistic rule must be met for accepting additionally mapped AAIGs to be restrained in the following round.

In the present implementation of the 4D-CHAINS algorithm, no mapping mistakes were made for the four protein targets. The NH-mapping coverage varied between 96 and 100% (Supplementary Figure 4; left column). To better evaluate the mapping performance of 4D-CHAINS, only the $^{13}$C–$^1$H correlated frequencies of α- and β-atoms were retained in the TOCSY input peak list to imitate the scenario of using a 4D CBHBCAHA(CO)NH experiment in conjunction with the 4D $^{13}$C,$^{15}$N edited HMQC-NOESY-HSQC. Interestingly, only the coverage dropped slightly but again no mistakes were introduced (Supplementary Figure 4; right column). This control experiment highlights the robustness of 4D-CHAINS that stems mainly from the predictive power of $^{13}$C–$^1$H correlated chemical shifts and the reliability of the connectivities established when carbon and proton frequencies are coupled.

For all AAIGs mapped to the protein sequence 4D-CHAINS obtains the assignment of aliphatic atoms by matching the observed $^{13}$C–$^1$H correlated frequencies to their distributions in the 2D probability density maps. First, the TOCSY frequencies are assigned to atom types of the previous amino acid in the sequence. Based on the amino acid type, pairs of $^{13}$C–$^1$H frequencies that differ by 0.2 p.p.m. or less in the carbon frequency are grouped to methylene moieties. The atom-type probability for these moieties is taken as the logarithmic average of the individual probabilities. Accordingly, all combinations of permutations are computed and the permutation with the highest probability provides the atom-type assignments for the TOCSY observed frequencies. It has been reported before[49] that automated assignments based on TOCSY-type transfer may interchange between atoms of certain amino acids because their chemical shift distributions overlap partially, as seen in the 2D probability density maps of correlated chemical shifts (Supplementary Figure 2). Another source of erroneous assignments may result from incomplete TOCSY patterns that become common as the size of the protein increases and also depend on the isotropic mixing period. For instance, in Leu residues, often the TOCSY observed correlations correspond to the α, β and one of the isopropyl atoms. Due to the missing correlations, any of the methyl groups could be wrongly assigned to the γ atom and vice versa, depending on the observed chemical shifts. 4D-CHAINS assigned all TOCSY correlations with an average error rate of 0.2% for 1845 $^{13}$C–$^1$H moieties in total (4 errors out of 1845 types) (Supplementary Figures 6 and 7). Minor TOCSY-based misassignments should have in principle little impact on structure calculations driven by long-range NOEs because they involve intraresidue atoms.

Next, all TOCSY-derived assignments are transferred to the NOESY spectra starting from the last residue and going backwards. For every amino acid, first the intraresidue NOE peaks are assigned ($i$) by matching the TOCSY assigned peaks of the successive residue, and then any sequential NOE peaks ($i$-1) by matching its own TOCSY assigned peaks. It has been noted from the early days of NMR[1], and supported later by inter-proton statistics[21], that for any given amide the observed NOE correlations to aliphatic protons are predominantly intraresidue ($i$) and sequential ($i$-1). As a proof of that, 4D-CHAINS traced 99% of TOCSY peaks as intraresidue NOE correlations and 89% as sequential NOE correlations. This analysis demonstrates that for any given amide and its successive one, a large portion of common NOEs they share correspond to the aliphatic atoms of the former. This is particularly true for the methyl groups that in principle yield strong NOE correlations to amides, both intraresidue and sequential. The only exception is the distant methyl group of Met. Intraresidue NOE correlations are uniformly present to any type of secondary structure (α-helix, β-sheet, loops), whereas sequential NOE correlations are most prevalent in β-sheets. Yet, most of the missing sequential NOE correlations correspond to certain atom types ($\delta$ and $\varepsilon$ of Lys, $\delta$ of Arg, $\varepsilon$ of Met, and to a lower extent $\gamma$ of Leu and $\gamma$1 of Ile).

Common NOEs are utilized to derive missing TOCSY-based assignments. For every residue separately, the NOESY peak intensities are normalized and peaks with low intensity (threshold 0.1 or specified otherwise) are left out. 4D-CHAINS scans the sequence backwards. For each residue where there is a missing assignment, it matches its unassigned NOE peaks with the unassigned NOE peaks of the next residue. For each peak where there is a match, a probability is derived for the missing atom-type assignments from the 2D density map of the particular residue type. Atom-type probabilities must belong to the 80th percentile of the density maps to be considered further. This filter prevents making any decisions when the correct assignment does not belong to any of the matched peaks. Accordingly, each probability is modified by the intensity of the corresponding peak to a score. This process is necessary to identify the correct assignment of methyl groups among the available options, because intraresidue methyl-amide NOEs yield stronger correlations. Several intensity transformations were tested extensively (see below) and the best performance for obtaining NOESY-type methyl assignments is given by the product of the 2D histogram probability and the intensity of the NOESY peak transformed by an exponential function, e.g., 2Dprob×(100×intensity$^2$). The highest score or product of scores provides the assignments for the missing atom types.

The efficiency of 4D-CHAINS in obtaining atom-type assignments from the 4D-HCNH NOESY spectrum has been tested in three different scenarios (Supplementary Figure 5 and Supplementary Table 1). In the current workflow 4D-CHAINS sought assignments not present in the TOCSY spectra. For the four different datasets, it assigned 13% of additionally assignable aliphatic atoms (277 carbon types) with an average error rate of 8.7% (24 errors out of 277 types). Then, it operated on synthetic data of a 4D CBHBCAHA(CO)NH experiment. It performed NH mapping successfully, assigned correctly all α- and β-atoms (1253 carbon types or 56.2% of all assignable atoms) and completed the missing assignments from the NOESY spectrum, where it assigned 37.9% of additionally assignable atoms (847 carbon types) with an average error rate of 5.1% (43 errors out of 847 types). Finally, only the backbone amide $^{15}$N,$^1$H HSQC frequencies were provided and 4D-CHAINS was asked to assign all aliphatic atoms from the 4D-HCNH NOESY spectrum (Supplementary Figure 8). 4D-CHAINS was able to assign 91.1% of all assignable aliphatic atoms (2033 carbon types) with an average error rate of 5.5% (112 errors out of 2033 types). In all cases the assignment error rate for methyl groups was lower: 7.9% for the first scenario (6 errors out of 76 methyls assigned), 3.8% for the second scenario (15 errors out of 393 methyls assigned) and 3.3% for the third scenario (15 errors out of 457 methyls assigned).

**Automated atom assignments using FLYA**. For FLYA resonance assignment calculations all available spectra were used as input, that is, 4D-HCNH TOCSY, 4D-HCNH NOESY and 4D-HCCH NOESY. The calculations were performed using the demo script provided with CYANA distribution shown in the Supplementary Methods.

**NOE assignment and structure determination using Rosetta**. NOE-based structural ensembles were generated using the csrosetta3 toolbox integrated within the Rosetta3 software suite. autoNOE-Rosetta[19] is one of the protocols included in the toolbox which performs automatic assignment of NOEs and structure determination based on the highly parallel RASREC-Rosetta[50] conformational sampling engine, which can successfully determine well-converged structures from sparse NMR data[19, 51]. The main principle is to iterate the NOE assignment algorithm alongside a multi-stage (I–VIII) conformational sampling process, towards obtaining a network of long-range NOEs that drive structure refinement to the global minimum of the Rosetta energy function[24]. The protocol uses as input initial assignments of NOE cross-peaks, derived from the chemical shift lists provided by 4D-CHAINS. The selection of high-ranking initial assignments of NOE cross-peaks depends on several factors, including: symmetry of the peaks, chemical shift matching score and network anchoring. Long-range NOE restraints and backbone chemical shift fragments guide the generation of batches of preliminary, low-resolution structures, that are in turn utilized to evaluate and refine the NOE assignments. Short and medium-range NOEs are also assigned by the program, but not utilized in structure refinement. The autoNOE-Rosetta further eliminates peaks during the sampling process. In the final stages, only highly converged, lowest-energy models that satisfy the maximum number of assigned NOE distance restraints are retained.

In practice, the process of setting up the protocol and generating models involves the following steps. (1) Preparation of chemical shift, NOE peaks and sequence files. From the 4D-CHAINS XEASY chemical shift table, we generate a TALOS[52] file and perform empirical prediction of backbone torsion angles using TALOS-N[53] for a given protein sequence. Based on the predicted chemical shift order parameter, we retain only the rigid regions of the structure. (2) Fragment selection[54] from high-resolution structures in the PDB[23]. We use the TALOS-N $\varphi$, $\psi$ and secondary structure-type predictions to bias the selection of 3- and 9-residue backbone fragments, excluding fragments derived from homologs to the target sequence present in the database. (3) Automated setup of autoNOE-Rosetta calculations for a range of restraint weight values.

According to this general procedure, we performed two sets of calculations for each target using NOE peak lists that included either (i) amide to aliphatic (HCNH) only, or (ii) amide to aliphatic and aliphatic to aliphatic (HCNH +HCCH). To improve sampling for nEIt, NOEs were supplemented by one RDC dataset. All calculations were setup with standard restraint weights of 5, 10, 25 and 50. The optimum restraint weight was selected based on an empirical cost function that considers individual restraint weights, Rosetta all-atom energies (talaris2014. wts[24]), and degree of structural convergence in each calculation. Finally, we select an ensemble of 10 lowest-energy structures that show minimum NOE violations. A detailed method to setup the calculations and analyze the models is available in the Supplementary Methods.

**NOE assignment and structure determination using CYANA**. The 3D structural ensembles of all four target proteins used in this study were calculated using the CYANA[27] software suite, supplied with the same input datasets as with autoNOE-Rosetta. Depending on the size of each protein target, CYANA calculations required 45–90 min on 4 CPUs. The script with all parameters for CYANA calculations is available in the Supplementary Methods.

**Ensemblator analysis**. Analysis of the ensembles was performed using the Ensemblator[25, 26] software for atom- and residue-level global and local comparisons. The Ensemblator first iteratively overlays pairs of structures and finally defines a "common core" of atoms that are consistently within a specified cutoff distance. For each set of comparisons, the needed cutoff distance was automatically determined by the Ensemblator to yield 20–40% of the atoms in the common core. These comparisons also yield pairwise weighted distance metrics that are used to embed the models into an $N$-dimensional space where $N$ is the number of models. Also, for any specified group of models, an exemplar was defined as the model having the shortest average distance to all other models in its own group. Global comparisons between groups are performed after the common core atoms are used to overlay structures and local backbone comparisons are calculated based on the locally overlaid dipeptide residual which converts $\varphi$, $\psi$ differences to a single distance[25]. The global and local comparisons involve quantifying the levels of variation for each residue within and between defined groups so that the level of intragroup variation can be compared with the intergroup variation.

Finally, the models in the crystallographic ensemble consisting of 51 aLP structures, used for Ensemblator analysis, were obtained from the CoDNaS[28] database by searching for α-lytic protease and utilizing all the available X-ray structures.

**Restraint violation analysis**. NOE restraint violations among the 10 lowest-energy models in each calculation are reported separately for different classes of

restraints assigned by autoNOE-Rosetta (Supplementary Tables 2, 6–9). First, restraints are automatically divided into three confidence classes according to a total assignment probability score[19]: highly confident (HI) (probability >70%), confident (MED) (probability >45%) and least confident (LOW) (probability <45%). Second, the ambiguity score reflecting assignment uniqueness[19] further classifies constraints into ambiguous (AMBIG) (ambiguity score >0.1), near unambiguous (NEAR_UNAMBIG) (ambiguity score <0.1) or unambiguous (UNAMBIG) (ambiguity score <0.01). Therefore, according to these criteria, each constraint can be classified into one of the six classes: HI_UNAMBIG, HI_NEAR_UNAMBIG, HI_AMBIG, MED_UNAMBIG, MED_AMBIG and LOW_AMBIG. Finally, due to the lack of stereo-specific assignments by 4D-CHAINS, the resulting autoNOE restraints are structurally degenerate and are therefore treated using an effective distance computed as the $r^{-6}$ average between all the possible pairs of atoms[55]. We used a 7 Å upper distance bound to identify violations in the resulting structurally degenerate NOE restraints, shown as an average over the 10 lowest-energy models in each structural ensemble. The choice of 7 Å as upper distance bound is attributed to the use of a 70 ms mixing time where through-space magnetization transfer between closer protons can happen within a maximum distance range of 7 Å[56]. This statement was found to be true by direct observation of distances corresponding to confidently assigned NOEs in the X-ray structure of aLP (PDB ID 1PO1).

**Computational cost**. 4D-CHAINS takes an average of half an hour to run on a commodity computer. All autoNOE-Rosetta structure calculations were performed at the UCSC Baker cluster with 13 compute nodes (AMD Opteron(tm), 2.4 GHz Processor 6378) and 32 cores per compute node. Typical message passing interface calculations are run in parallel on 100 cores, and depending on target size, take an average of: (i) 6–8 h (150 aa), (ii) 12–14 h (200 aa) and (iii) 16–18 h (250 aa). A total of approximately 2 million CPU hours was used for the various development stages of the method.

**CS-Rosetta support for NMR Exchange Format**. NMR restraint datasets are now represented using a new open standard, NMR Exchange Format (NEF)[57]. NEF is a self-contained format designed to be machine readable by common NMR structure determination software tools. The file is divided into sections where each section corresponds to the data used for structure calculation. Full specification of each section in NEF can be found at: https://github.com/NMRExchangeFormat/NEF/blob/master/specification/Overview.md.

NEF provides support for a set of identifiers to be used by software tools. We utilize the identifiers provided by the NEF specifications to design NEF converter and NEF parser tools as part of the csrosetta3 toolbox. NEF converter is a tool that can take sequence information, chemical shift assignments, RDC data (if available), distance restraints and peak information used for structure calculation and convert it to the standard NEF file format for deposition in databases (that support NEF).

Similarly, we also provide a series of tools to extract respective information from NEF file into FASTA, NOE restraint, chemical shift and peak files for subsequent automatic setup of CS-Rosetta for structure calculations. See supplementary information for detailed commands to convert to NEF and parse NEF file.

**wwPDB data deposition**. Currently, wwPDB does not support the NEF format, and we therefore utilized the NEF to BMRB translator program provided by BMRB[58] (https://github.com/kumar-physics/BMRBTranslator) to convert from NEF to NMR-STAR format for data deposition. The deposited NMR-STAR file consists of chemical shifts, peaks and RDCs used for structure calculation.

**Code availability**. 4D-CHAINS is available on github (https://github.com/tevang/4D-CHAINS) for non-commercial usage. The updated CS-Rosetta (version 3.4) software and the detailed documentation for installation and usage can be obtained at the CS-Rosetta web server (https://csrosetta.chemistry.ucsc.edu). The current version (3.4) of CS-Rosetta also supports conversion of data to NMR Exchange Format for deposition to the wwPDB.

**Data availability**. Biological Magnetic Resonance Bank: chemical shifts, peak lists, RDCs have been deposited under 30322, 30325, 30326, and 30327 BMRB codes. Protein Data Bank: restraint lists and coordinates have been deposited under 5WOT, 5WOX, 5WOY, and 5WOZ PDB codes. Other data are available from the corresponding authors upon reasonable request.

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

## Acknowledgements

We thank Peter Lukavsky, Richard Stefl and Arie Geerlof for providing NMR samples and Alison Barrett for assistance during the early stages of developing the csrosetta3 code. This research was supported by a grant from the Czech Science Foundation (15-22380Y), project CEITEC 2020 (LQ1601) with financial contribution from the MEYS CR and National Programme for Sustainability II, and a Marie Curie Career Integration Grant (618223) to K.T., by NIH grant R01GM083136 to P.A.K., a K-22 Career Development and an R35 Outstanding Investigator Award to N.G.S. through NIAID(AI2573-

01) and NIGMS(1R35GM125034-01), respectively. CIISB research infrastructure project LM2015043 funded by MEYS CR is gratefully acknowledged for the financial support of the measurements at CEITEC Josef Dadok National NMR Centre. We acknowledge the UCSC 800 MHz NMR facility supported by the Office of the Director, NIH, under High End Instrumentation (HIE) Grant S10OD018455.

## Author contributions

K.T. conceived and together with N.G.S. designed the project. T.E. and K.T. developed and executed the 4D-CHAINS algorithm. J.N. and K.T. developed, recorded and analyzed 4D NMR experiments. S.N. and N.G.S. developed the CS-Rosetta3 software and performed parallel structure calculations, structure analysis and validation, additional testing of 4D-CHAINS and NMR experiments. A.E.B. and P.A.K. performed Ensemblator analysis. R.R.D. and V.V. prepared labeled protein samples and recorded RDC data for nEIt. N.G.S., S.N., T.E. and K.T. wrote the paper, with feedback from all authors.

## Additional information

**Competing interests:** The authors declare no competing financial interests.

