## [Peer Review File · Nature Communications]

Reviewers' comments:

Reviewer #1 (Remarks to the Author):

Evangelidis and coworkers present a novel computational approach called 4D-CHAINS aimed at automatically solving protein structures using two 4-dimensional spectra in an 'unsupervised' manner and with minimal expert data analysis. The claim is validated via a set of four protein solidly positioned in the medium size range of complexity (15-27 kDa).

The methodology adds to the well-established ROSETTA arsenal of NMR structure determination protocols that has been under development in the last decade. The novelty here is the introduction of Amino Acid Index Groups (AAIGs) and Overlap Layout Consensus (OLC) as a way of finger printing the amino acids, assigning and then positioning them along the protein backbone. The algorithm goal is achieved automatically in a single step using a combination of 4D backbone TOCSY and NOESY of the H-C-NH-type using doubly labeled samples. The result is significant in the field.

The testing is conducted with a variety of samples that have distinct secondary structures and tertiary folds from helical bundles to beta-barrel to 'stress' the program and give a likely usage scenario in the field. The results are convincing from the standpoint of structure quality and their proximity to a 'native fold' represented by the crystal structure. The work in that regard is comprehensive and the validation does indicate that the software performs as intended up to ~30 kDa.

Strategies aimed at improving the sidechain assignment problem are worthy, the challenge stems from both resonance overlap and unfavorable NMR relaxation properties. Both aspects become progressively worse with increasing size as the number of overlapping residues increases and the protein tumbles more slowly. A solution is to thin out the observable spins with deuteration and in so doing reducing overlap and alleviating the relaxation issues particularly at the CA-HA position. This preamble brings me to the main criticism of the work and the claim that the method works for large proteins. Of the vast array of NMR experiments available for residue assignment and structure determination using protonated samples, running a 4D amide-observed TOCSY experiment would be at the bottom of my list knowing it is among of the most insensitive experiment available. Even with the short TOCSY spinlock time used (12ms) the magnetization losses are substantial. The authors employ 850-950MHZ state of the art instrumentation and cryoprobes to achieve such results up to a 27 kDa protein. In terms of complexity that result would be considered underwhelming by today's standards.

Furthermore, the authors achieve ~80% overall assignments from the 4D-CHAINS but contribution from having additional assignment vs having just methyl and perhaps aromatics is not clear to me. From that stand point the work represent a move sideways more than forward.

To the best of my knowledge the 4D HC(CC-TOCSY(CO))NH experiment presented by Mobli et al in 2010 J Magn Res was demonstrated on a small peptide and has so far been used for peptides and IDPs. In addition, this experiment would not achieve assignment of aromatic residues. That could explain the poor results in cyana from the HNCH NOEs-only runs.

The 4D experiments are 4 days each (x2 or x3) even with non-uniform sampling bringing the experimental time close to traditional acquisition but without the sensitivity of the ^{13}C -edited 3D HCCH-COSY or TOCSY.

The strategy is very well suited for midsize proteins, or for well-behaved genomics targets that are soluble, stable and homogeneous in linewidth as stated in the summary but not for large proteins as stated in the title. The point should be clarified for publication. The manuscript is otherwise very detailed, well written and the should be well received by the target audience in biophysics.

Major:

1-It is not clear to me how much of the results are from using two backbone-based 4Ds and what is the contribution from the HCCH-type 4Ds.

2-The title should be revised stressing the fully automated vs. the large size which is misleading given

the size of the proteins in the study.

3-For the largest target, the s/n of the 4D proposed experiment should be compared to the traditional 3D HCCCO-NH TOCSY. It should be demonstrated that even the case of the largest target sufficient s/n can be achieved.

Minor:

4-Compare the number of raw NOEs that can be extracted from this vs traditional assignment.

5-Which target was used for the Table S1 and how is the error rate affecting the outcome? 15% seems high indicating the NMR restraints are loosely enforced by the protocol that relies mostly on Rosetta force field.

6-how much sidechain assignment is in fact required to achieve correct native fold?

7-how many critical methyl NOE are identified vs the manual assignment and how important are they?

This ties into the idea that the methyl and amides are the most important moieties for folding. Are the aromatic sidechains assigned? Only alpha and betas appear in the map in Fig S3.

Reviewer #2 (Remarks to the Author):

The paper of Evangelidis et al. presents an interesting study about automated structure determination of protein by solution state NMR spectroscopy. As the authors mentioned in the introduction this is a very active field of research where improvement need to be made and has great potential in structural biology. They have shown on four different proteins that the method is capable to provide quickly with medium accuracy (2 Angstrom) the structure of the proteins.

The article is well written (abstract, introduction and conclusion are clear and accessible), the data are well presented (with minor comments, see below) and the methodology adequate. The references are appropriate.

As mentioned, previous version of Rosetta can perform structure determination of protein solely based on chemical shift up to 12kDa and up to 25 kDa for the (RASREC)-Rosetta using backbone RDCs and amide NOEs.

In the presented article, the authors claim that autoNOE-Rosetta will benefit from the new assignment protocol using 4D NMR experiments. This new protocol (4D-CHAINS) is the main contribution of the authors. It is well documented and integrated with the autoNOE-Rosetta pipeline, making the method accessible to the whole community.

Even if the size of the largest protein does not exceed significantly 25 kDa, 4D-CHAIN allows the use of direct measured experimental data (long distance information) instead of a repository or sparse experimental data. It will lead to higher quality of the structure predictions. This milestone is crucial for automatic NMR structure calculation to be widely spread.

I have few comments and concerns:

1. Very important. These methodologies rely nearly exclusively on good peak list inputs. In the current form, the peak lists are given to the program as input. In order to make the protocol truly unsupervised and automated the lists should be generated by a computer as well, using a peak picking program uniformly for all four examples (namely, no manual curation). The 4D spectra are in that case advantageous because they reduce the possible overlaps that could happen in 3D spectra. The authors should run their analysis with the new inputs lists and report their results.

2. In figure 2d. It is my understanding that the data used for that system are complemented by RDCs. This should appear on the figure or in the caption otherwise it is misleading (even if it is stated in table 1 and in the text). The reader could think that only NOEs data are used as for the other three proteins. The structure found without using RDCs should be reported in the supplementary information.

3. In figure 2. The RASREC predictions should be reported in the supplementary information and compared to the reference (Xray) structures as well as the reported predictions.
4. In table 1. The authors should comment why the fraction of converged residues is better for the unsupervised calculations compare to the supervised calculations.
5. In supplementary 1, add a panel that shows the NOESY and TOCSY connections between the atoms of the protein (residues i and i-1), following the description of the two 4D spectra.
6. Important. The probability density maps represent the chemical shift from the majority of proteins. What would be the outcome of the protein structure prediction if the shifts of a certain protein region would be far from the expected values and randomly shifted. In short, the conditional probabilities for these residues will drop severely toward zero. This would be the case if a small molecule containing aromatic groups would bind the protein, as the authors suggest in their conclusion.
7. Supplementary Figure 4 shows spectra with a Sweep Width larger than 60 ppm while the method report 8000Hz SW. Please comment, correct.
8. Supplementary Figure 7. It is my understanding that the case "NOESY" is actually using known and fixed backbone assignments. If this is the case, it should appear on the figure.
9. Supplementary Figure 8. The figure is too small to be read properly.

Reviewer #3 (Remarks to the Author):

This paper reports a new algorithm for automated NMR resonance assignments using three NMR peak-lists. The algorithm relies (i) peak-patterns provided by a 4D TOCSY spectra for residue typing (ii) intra-residues and sequential connectivities present in 4D TOCSY and 4D NOESY spectra and (iii) expected peak positions for chemical shifts distribution. It has been applied on 4 large protein targets (from 15 to 27kDa) for which the required spectra have been recorded. The algorithm yields near complete and accurate assignments (94% of aliphatic shifts) when compared to assignments curated by a trained user on the basis of the algorithm's output. In a second stage, the generated assignment lists were combined with two 4D NOESY peak-lists for structure determination using the autoNOE-Rosetta software. As a result, generated models show a large degree of convergence. Models generated from curated or automated assignment are very similar in terms of convergence and assigned long-range NOE restraints. For the only target for which an X-ray structure is also available, it is shown that structure accuracy can be improved to 1.4Å RMSD when automatically assigned resonances from TOCSY-NOESY peak-lists are combined with restraints from two 4D NOESY peak-lists.

The whole procedure (types of spectra recorded, algorithm for automated assignment and structure calculation procedure) described in the paper is original, very efficient and potentially of great use for the NMR community since human intervention is greatly limited and the time to collect and analyze the data is drastically reduced compared to standard approaches commonly used by the community.

However, it is difficult to really assess the necessity of the approach since the results are not put in perspective. It is a method paper, so the performance must be compared to state of the art methods for automated resonance assignment and structure calculation from NOE peak-lists, e.g.:

- The FLYA algorithm (Schmidt & Guntert, 2012) is to date the most efficient tool for backbone and side-chain resonance assignment. While it is briefly mentioned in the introduction, no comparison is shown on the respective performance of FLYA and 4D CHAINS with the same input data. It can be that the algorithm presented is the only one that can do the job, but we have no idea of that.

- CYANA software (Guntert's group) is by far the most used approach by the community for structure calculation from NOESY spectra. There should a thorough comparison between CYANA and autoNOE-rosetta so that potential users know if it is worth resorting to the CPU-demanding Rosetta suite.

Other remarks

- The title sounds a bit pompous and could be tuned down:

* "fully automated": given the lengthy instructions given at the end of the supplementary materials on how to run CS-rosetta after 4D-CHAINS, it would be more fair to discard the "fully"

* "from 2 spectra": it's actually more from 4 spectra (HSQC, TOCSY, 2x NOESY) plus RDC for the 27kDa target. Maybe put instead "from a limited number of spectra"

- Details must be given on the peak-picking procedure (manual, automated then curated or fully automated ?). Also, the sensitivity to missing peaks should be discuss further.

- Introduction : ARIA do not assign resonances (only NOE peak-lists)

- RPF analysis (from Monteliones's group) provides standardized measures to assess the reliability of an NMR model. In particular, DP-score is known to correlate very well with structure accuracy. The authors should present DP-scores for the different targets.

- page 6: CYANA has been used with identical data as autoNOE-Rosetta for the aLP target. I don't see any description of the CYANA setup used.

- Conclusion: the reference to high-throughput structural genomics projects is outdated (Heineman et al. 2001).

- Methods: "restraint violations analysis" and Supp. Table 4: A uniform upper bound of 7A is used to measure violation of NOE restraints. What is the rationale for that ?

Point-by-point response

We would like to thank all the reviewers for their comments. We believe that the overall quality of the revised manuscript has been further improved as a result of their constructive feedback. Below are our responses addressing the points raised. Here, we refer to the additional table and figure numbers as they appear in the revised manuscript.

Reviewer #1

Evangelidis and coworkers present a novel computational approach called 4D-CHAINS aimed at automatically solving protein structures using two 4-dimensional spectra in an ‘unsupervised’ manner and with minimal expert data analysis. The claim is validated via a set of four protein solidly positioned in the medium size range of complexity (15-27 kDa).

The methodology adds to the well-established ROSETTA arsenal of NMR structure determination protocols that has been under development in the last decade. The novelty here is the introduction of Amino Acid Index Groups (AAIGs) and Overlap Layout Consensus (OLC) as a way of finger printing the amino acids, assigning and then positioning them along the protein backbone. The algorithm goal is achieved automatically in a single step using a combination of 4D backbone TOCSY and NOESY of the H-C-NH-type using doubly labeled samples. The result is significant in the field.

The testing is conducted with a variety of samples that have distinct secondary structures and tertiary folds from helical bundles to beta-barrel to ‘stress’ the program and give a likely usage scenario in the field. The results are convincing from the standpoint of structure quality and their proximity to a ‘native fold’ represented by the crystal structure. The work in that regard is comprehensive and the validation does indicate that the software performs as intended up to ~30 kDa.

We thank the reviewer for her/his comments.

Strategies aimed at improving the sidechain assignment problem are worthy, the challenge stems from both resonance overlap and unfavorable NMR relaxation properties. Both aspects become progressively worse with increasing size as the number of overlapping residues increases and the protein tumbles more slowly. A solution is to thin out the observable spins with deuteration and in so doing reducing overlap and alleviating the relaxation issues particularly at the CA-HA position. This preamble brings me to the main criticism of the work and the claim that the method works for large proteins. Of the vast array of NMR experiments available for residue assignment and structure determination using protonated samples, running a 4D amide-observed TOCSY experiment would be at the bottom of my list knowing it is among of the most insensitive experiment available. Even with the short TOCSY spinlock time used (12ms) the magnetization losses are substantial. The authors employ 850-950MHZ state of the art instrumentation and cryoprobes to achieve such results up to a 27 kDa protein. In terms of complexity that result would be considered underwhelming by today’s standards.

We share the reviewer’s concern and are aware of the TOCSY limitation related to protein size and tumbling time. As a rule of thumb, we recommend recording the $^{15}\text{N}/^1\text{H}$ 2D plane of the experiment using a full (incremental) sampling list first. Our methodology is applicable if the number of observed signals in the 2D plane are $\geq 50\%$ of expected, based on a standard 2D $^{15}\text{N}/^1\text{H}$ HSQC experiment recorded with the same number of points. It is further worth noting that, during the initial stages of development of our method, we tested three different TOCSY mixing times, 12, 18, and 24 ms, and found that the shorter mixing time yields the best overall S/N. Therefore, we used 12 ms as the default for all the protein targets. Finally, even if the TOCSY patterns are incomplete due to relaxation losses, additional assignments can be derived from the complementary 4D NOESY experiment, which has higher sensitivity and gives correlations for the majority of observable nuclei in our target set. These important points have been highlighted in paragraph 1 under NMR data collection subsection on page 15 of the revised manuscript.

Furthermore, the authors achieve ~80% overall assignments from the 4D-CHAINS but contribution

from having additional assignment vs having just methyl and perhaps aromatics is not clear to me. From that stand point the work represent a move sideways more than forward.

We apologize for not being sufficiently clear about the practical utility of different assigned groups in providing structural constraints. To be more specific, 80% of the assignments result from the TOCSY-NOESY *combination*. First, the TOCSY patterns are easy to interpret when carbon and proton frequencies are correlated, which in turn, helps us to obtain assignments that are almost error-free. Second, the assignments missing from TOCSY are obtained from the complementary NOESY spectrum, which allows an overall completeness of 95% with an error rate of just 1.5%. As shown in the new **Supplementary Table 2** on page 24 of the Supporting Information document, the vast majority of methyl atoms are indeed correctly assigned. To this extent, the use of 4D-CHAINS greatly simplifies the methyl assignment process relative to more traditional approaches using site-specific labelling of select amino acid groups (such as methyl groups and aromatics) and typically require two different labelling schemes for assignments and NOE measurements. Finally, we performed several structure calculations using autoNOE-Rosetta and demonstrated that the missing aromatic assignments do not compromise the quality of the resulting structures for all targets tested here. Overall, our combination of 4D-CHAINS with autoNOE-Rosetta provides a robust alternative approach leveraging a highly automated process to obtain reliable structures in minimal time. The important points discussed here have been updated in the revised manuscript (paragraph 1/page 8).

To the best of my knowledge the 4D HC(CC-TOCSY(CO))NH experiment presented by Mobli et al in 2010 J Magn Res was demonstrated on a small peptide and has so far been used for peptides and IDPs. In addition, this experiment would not achieve assignment of aromatic residues. That could explain the poor results in cyana from the HNCH NOEs-only runs.

Indeed, the application of 4D TOCSY experiments on larger molecules has not been previously demonstrated in the literature. Here, our results show that 4D TOCSY is suitable for proteins in the ~25 kDa range. To make this point clearer, we have added a sentence in the manuscript stating that “The largest protein target of size 27.3 kDa (Enzyme I) was chosen based on its apparent correlation time of ~15 ns that still allows for TOCSY transfer to occur (**Supplementary Fig. 2**)”, in paragraph 2/page 3 of the revised manuscript. Also, while 4D-CHAINS does not provide aromatic assignments, this has a minimal impact on autoNOE-Rosetta structure calculations, primarily due to the use of a high-resolution energy function to model the sidechain conformations. This is in sharp contrast with CYANA, which requires a higher coverage of assignments. As expected, CYANA performs better with manual assignments which contain most observed aromatic residues.

The 4D experiments are 4 days each (x2 or x3) even with non-uniform sampling bringing the experimental time close to traditional acquisition but without the sensitivity of the ^{13}C –edited 3D HCCH-COSY or TOCSY.

We share the reviewer’s concerns regarding the loss of sensitivity in our proposed 4D experimental strategy, and further agree that the experimental time needed to acquire three 4D NUS spectra is comparable to the total acquisition time of several conventional 3D experiments. However, the analysis of 3D experiments is laborious and further complicated by resonance overlap, which becomes more pronounced with increasing target size. Thus, from the user’s standpoint, it is preferable to operate using a pair of complementary experiments which yield the same information in a higher-dimensionality dataset. Furthermore, the computational identification of AAIGs is greatly enhanced by the availability of correlated $^{13}\text{C}/^1\text{H}$ frequencies for all sidechain correlations, that can be readily obtained from our 4D experiments. Finally, as we show in our benchmark data, the $\sqrt{2}$ losses in sensitivity for the 4th indirect dimension are not prohibitive for highly concentrated samples of stable proteins, which can still yield very rich datasets allowing accurate structure determination. These key points have been added in the revised text (paragraph 1/page 16).

The strategy is very well suited for midsize proteins, or for well-behaved genomics targets that are soluble, stable and homogeneous in linewidth as stated in the summary but not for large proteins as stated in the title. The point should be clarified for publication. The manuscript is otherwise very detailed, well written and the should be well received by the target audience in biophysics.

This point has been clarified at several points in discussion. We would like to thank the reviewer again for providing constructive feedback that has helped improve the quality of our manuscript.

Major:

1-It is not clear to me how much of the results are from using two backbone-based 4Ds and what is the contribution from the HCCH-type 4Ds.

We thank the reviewer for bringing up this point. For sidechain chemical shift resonance assignments 4D-CHAINS uses only two spectra, a 4D-HCNH TOCSY and 4D-HCNH NOESY. The third spectrum, HCCH NOESY, is utilized to obtain additional distance restraints during autoNOE-Rosetta structure calculations. This point has been highlighted in the revised manuscript (paragraph 2/page 4).

2-The title should be revised stressing the fully automated vs. the large size which is misleading given the size of the proteins in the study.

Our title has now been changed to: “Automated NMR resonance assignments and structure determination using a minimal set of 4D spectra”

3-For the largest target, the s/n of the 4D proposed experiment should be compared to the traditional 3D HCCCO-NH TOCSY. It should be demonstrated that even the case of the largest target sufficient s/n can be achieved.

The magnetization transfer pathway in the 4D HC(CC-TOCSY(CO))NH is the same as in the case of standard 3D HCCCO-NH TOCSY experiments. Therefore, the 4D HC(CC-TOCSY(CO))NH experiment used here in principle should not suffer from lower sensitivity relative to a standard 3D HCCCO-NH TOCSY within individual 2D planes. To test this, we have acquired both the 2D $H^{\text{aliphatic}}-H^{\text{N}}$ and $C^{\text{aliphatic}}-H^{\text{N}}$ planes in either 4D HC(CC-TOCSY(CO))NH and conventional 3D HCCCO-NH TOCSY experiments for the largest target, nE1 of 27.3 kDa. The new **Supplementary Fig. 2** shows the overlay of the 2D spectra measured using our 4D and standard 3D pulse sequences, respectively, demonstrating similar sensitivity in both experiments. The 1D projections show ~5-10% lower sensitivity for some signals in the 4D experiment, which we attribute to different CC-TOCSY spin lock sequence used (FLOPSY-16 in the 4D vs a shorter DIPSI-2 sequence in the 3D). While in the final 4D datasets, ^{13}C and ^1H T_2 relaxation during the extra (short) chemical shift evolution step will contribute to an additional loss of sensitivity, the resulting datasets are of sufficient s/n to observe a very large percentage of ^{13}C - ^1H correlations, as shown in our supporting data (**Supplementary Fig. 5**, **Supplementary Fig. 8**, **Supplementary Fig. 9**).

Minor:

4-Compare the number of raw NOEs that can be extracted from this vs traditional assignment.

We thank the reviewer for his/her suggestion. We have provided a detailed comparison of raw, long-range NOEs identified by autoNOE-Rosetta and those assigned by the traditional method, CYANA in paragraph 2 on page 7 of the revised manuscript and in **Supplementary Figs. 16** and **17** on pages 20 and 21 of the Supporting Information document.

5-Which target was used for the Table S1 and how is the error rate affecting the outcome? 15% seems high indicating the NMR restraints are loosely enforced by the protocol that relies mostly on Rosetta force field.

Supplementary Table 1 lists the performance of 4D-CHAINS in obtaining NOESY-based assignments for all protein targets. The performance is assessed using only 2D probability heat maps [top table labelled: Using 2Dprob] or a combined function that takes into account the corresponding relative peak intensities [bottom table labelled: Using 2Dprob * (100 * intensity²)]. Here, the total number of aliphatic carbons that can be assigned is 2,232. For the TOCSY-NOESY setting, 1,845 assignments are initially derived from the TOCSY spectrum with an error rate of 0.2%. In a following step, 4D-CHAINS analyzes common NOEs between successive residues to identify any of the missing 387 carbon types towards increasing the overall assignment completeness. When it considers only the conditional probabilities among the common NOEs, it assigns 278 additional carbon types with an error rate of 15.8% for the NOESY-based assignments (44 out of 278). When it considers the product of the 2D-histogram probability and the intensity of the NOESY peak, it assigns 277 additional carbon types, with an error rate of 8.7% (24 out of 277). Therefore, this is the default setting implemented in

4D-CHAINS to yield 94% assignment coverage with a combined error rate of 1.3% (TOCSY-based assignments: 1841 correct, 4 wrong. NOESY-based assignments: 253 correct, 24 wrong). The main conclusion here is that peak intensities improve 4D-CHAINS performance in obtaining NOESY-based assignments by increasing the overall assignment completeness without significantly compromising the overall level of correctness. The structural models derived from Rosetta show that a combined error rate below 2% does not affect the quality of the structures when compared to the supervised, error-free assignments. This point has been clarified in the Table legend on page 23 of the Supporting Information document.

6-how much sidechain assignment is in fact required to achieve correct native fold?

The use of autoNOE-Rosetta allows us to obtain accurate protein models demonstrating the correct fold from low as 60-70%. This is based on benchmark calculations performed by randomly removing entries from our “best effort” supervised assignment lists for target aLP. This important point has been added to the revised manuscript (paragraph 2/page 5).

7-how many critical methyl NOE are identified vs the manual assignment and how important are they? This ties into the idea that the methyl and amides are the most important moieties for folding. Are the aromatic sidechains assigned? Only alpha and betas appear in the map in Fig S3.

Indeed, the methyls provide many important structural constraints. Specifically, we found that ~25% of all NOEs identified by autoNOE-Rosetta correspond to methyl-methyl connectivities. The methyl NOEs identified are distributed across the entire protein for all the targets under study. The total numbers of methyl NOEs identified for both supervised and automated assignments have been reported in **Supplementary Table 5** on page 27 of the Supporting Information document and the corresponding text has been updated in paragraph 1 on page 8 of the revised manuscript. Additionally, **Supplementary Fig. 4** (previously Supplementary Figure 3) shows the assignment maps for the aliphatic moieties of every amino acid. Aromatic side chains are not displayed because 4D-CHAINS does not consider aromatic frequencies.

Reviewer #2

The paper of Evangelidis et al. presents an interesting study about automated structure determination of protein by solution state NMR spectroscopy. As the authors mentioned in the introduction this is a very active field of research where improvement need to be made and has great potential in structural biology. They have shown on four different proteins that the method is capable to provide quickly with medium accuracy (2 Angstrom) the structure of the proteins.

The article is well written (abstract, introduction and conclusion are clear and accessible), the data are well presented (with minor comments, see below) and the methodology adequate. The references are appropriate.

As mentioned, previous version of Rosetta can perform structure determination of protein solely based on chemical shift up to 12kDa and up to 25 kDa for the (RASREC)-Rosetta using backbone RDCs and amide NOEs.

In the presented article, the authors claim that autoNOE-Rosetta will benefit from the new assignment protocol using 4D NMR experiments. This new protocol (4D-CHAINS) is the main contribution of the authors. It is well documented and integrated with the autoNOE-Rosetta pipeline, making the method accessible to the whole community.

Even if the size of the largest protein does not exceed significantly 25 kDa, 4D-CHAIN allows the use of direct measured experimental data (long distance information) instead of a repository or sparse experimental data. It will lead to higher quality of the structure predictions. This milestone is crucial for automatic NMR structure calculation to be widely spread.

We thank the reviewer for her/his kind remarks on the novelty and practical utility of our method.

I have few comments and concerns:

1. Very important. These methodologies rely nearly exclusively on good peak list inputs. In the current form, the peak lists are given to the program as input. In order to make the protocol truly unsupervised and automated the lists should be generated by a computer as well, using a peak picking program uniformly for all four examples (namely, no manual curation). The 4D spectra are in that case advantageous because they reduce the possible overlaps that could happen in 3D spectra. The authors should run their analysis with the new inputs lists and report their results.

We acknowledge that automated pick peaking is the major drawback of all approaches aiming to automate chemical shift assignments. In principle, peak picking methods should perform much better on 4D spectra, since peak overlap is almost absent. A full evaluation of the performance of different peak picking algorithms on 4D data, although very important, falls outside of the scope of our work which introduces key innovations in assignment strategy and structure calculations to promote a fully automated pipeline from the point of input peak lists to structure. Thus, to avoid any confusion on the main claims of our work, we have removed the phrase “fully automated” from the title and abstract.

Preliminary evaluation of the SPARKY built-in picker shows a potential challenge, where a small number of additional artifact peaks are picked in the vicinity of strong peaks. In the case of TOCSY data, any additional incorrect peaks would negatively impact the amino acid type predictions by CHAINS, and leads to the requirement of manual curation of the resulting SPARKY lists in a preceding step towards removing the artifacts. In practice, our 4D-based strategy makes any manual aspects of the peak curation process very efficient from a user’s perspective. Here, the 2D reference HSQC experiments can be used to further facilitate and automate the process by enabling a restricted peak picking approach. This is highlighted in our revised manuscript (Methods) as follows: “Peaks were picked automated and curated manually. First the 4D-NOESY spectrum was picked at a user defined noise level using both $^{15}\text{N}, ^1\text{H}$ - and $^{13}\text{C}, ^1\text{H}$ -HSQC peaks as filters. Then the 4D-TOCSY spectrum was picked using the 4D-NOESY peaks as filters. Accordingly, all planes were inspected side by side in both spectra and picked artefacts were removed. This is a very fast procedure because the spectra can be synchronized since they share all four dimensions.”

2. In figure 2d. It is my understanding that the data used for that system are complemented by RDCs. This should appear on the figure or in the caption otherwise it is misleading (even if it is stated in table 1 and in the text). The reader could think that only NOEs data are used as for the other three proteins. The structure found without using RDCs should be reported in the supplementary information.

That is correct, the Enzyme I structures reported in **Figure 2d** were computed using RDCs along with chemical shifts and NOEs. To make this point clearer, the use of RDC data in **Figure 2d** has been explicitly stated at the legend (page 13). Additionally, the structures along with convergence statistics for Enzyme I without using RDC data are now reported in a new **Supplementary Fig. 18** on page 22 of Supporting Information document, and the corresponding text referring to **Supplementary Fig. 18** has been updated in paragraph 1 on page 8 of the revised manuscript.

3. In figure 2. The RASREC predictions should be reported in the supplementary information and compared to the reference (Xray) structures as well as the reported predictions.

A thorough comparison between RASREC and autoNOE-Rosetta predictions with respect to reference structures has been reported in a new **Supplementary Fig. 15** on page 19 of Supporting Information document. These results are outlined in detail in paragraph 2/ page 7 of the revised manuscript.

4. In table 1. The authors should comment why the fraction of converged residues is better for the unsupervised calculations compare to the supervised calculations.

In the case of Enzyme I, we observe that convergence statistics using the automated assignments are indeed higher. After further analysis of our calculations, we attribute this effect to the identification of the correct fold during the early stages of the autoNOE-Rosetta structure calculation process. For all other targets, we observe the expected behavior, *i.e.* that the supervised assignments perform slightly better than the automated process. Paragraph 1/ page 8 has been revised to reflect this observation.

5. In supplementary 1, add a panel that shows the NOESY and TOCSY connections between the atoms of the protein (residues *i* and *i-1*), following the description of the two 4D spectra.

We thank the reviewer for his/her suggestion. The panel showing 4D-HCNH NOESY and 4D-HCNH TOCSY connections between atoms of consecutive residues in the protein have been added in **Supplementary Fig. 1** on page 1 of the Supporting Information document.

6. Important. The probability density maps represent the chemical shift from the majority of proteins. What would be the outcome of the protein structure prediction if the shifts of a certain protein region would be far from the expected values and randomly shifted. In short, the conditional probabilities for these residues will drop severely toward zero. This would be the case if a small molecule containing aromatic groups would bind the protein, as the authors suggest in their conclusion.

We share the reviewer's skepticism, and address it as follows: VASCO contains many curated entries (4270 on average for each amino acid type), which is sufficiently large to allow reconstruction of valid 2D correlated C-H probability distributions (2D probability histograms). These correlated 2D histograms were smoothed using a Gaussian kernel function in order to expand the probability density to neighboring zero-probability regions. Such regions include some of the upfield chemical shift values observed in the presence of ring current effects or aromatic ligand binding. Hence, the conditional probability for that C-H pair type remains >0 . Notably, the C-H pair types that experience more intensely the upfield chemical shift effect are those at the end of the aliphatic chain. Thereby, when the correlated 2D histogram probability of a particular C-H pair is zero while the probability of at least one other C-H pair for that amino acid (e.g. CA-HA) is non-zero, 4D-CHAINS calculates conditional probabilities by considering only the smoothed 1D Carbon probability, because carbon is less susceptible to shifts inflicted by its chemical environment (i.e. the shift of the carbon relative to its average value in ppm is lower than the relative shift of the proton). Moreover, after smoothing with a Gaussian kernel, the 1D carbon probability distribution covers extreme ppm values and hence is never zero. Although we haven't encountered cases where all conditional probabilities for a given spin system are close to zero, we acknowledge this as a possible event. In this case no predictions would be made by 4D-CHAINS. The spin system would not be part in any of the resulting assignment solutions, instead it would lead to a gap in the protein sequence. This is still not detrimental to our approach, since *Rosetta* can reliably handle individual gaps of 3-9 residues long during the structure calculation and loop modeling process using a sequence bias to select native-like fragments from the PDB. Finally, it is worth noting that the longest gap we observed in our benchmark set of 4 targets with 725 residues in total was of length 5 (for target aLP) and this had a negligible impact on the local quality of the final *Rosetta* structures.

7. Supplementary Figure 4 shows spectra with a Sweep Width larger than 60 ppm while the method report 8000Hz SW. Please comment, correct.

Thank you very much for noticing the discrepancy. The sweep width in all carbon dimensions is 8000Hz with the carrier centered in the middle of the carbon frequency range (39 or 40 p.p.m.). Therefore, several signals are folded in the carbon dimension. For the purpose of visual aid in **Supplementary Fig. 5** (previously Supplementary Fig. 4), all folded peaks have been unfolded manually to make them appear as being in the true frequency range with respect to rest of the peaks. The same reason applies to data presented in **Supplementary Fig. 11** (previously Supplementary Fig. 10) that showcases the overlays of experimental data with the density map. For clarification, we have added the following sentence to both **Supplementary Figs 5 and 11** legends: "For visualization purposes, the carbon frequencies of folded peaks have been unfolded manually", on pages 7 and 15 of the Supporting Information document.

8. Supplementary Figure 7. It is my understanding that the case "NOESY" is actually using known and fixed backbone assignments. If this is the case, it should appear on the figure.

The case "NOESY" uses as input known and fixed $^1\text{H}, ^{15}\text{N}$ -HSQC assignments. All aliphatic assignments (including $\text{C}\alpha$ - $\text{H}\alpha$, $\text{C}\beta$ - $\text{H}\beta$) are derived exclusively from the 4D-HCNH NOESY spectrum. **Supplementary Fig. 8** (previously Supplementary Fig. 7) legend has been updated to explain in detail the settings listed in the graphs "TOCSY-NOESY", "CAB-NOESY", "NOESY", and "FLYA".

9. Supplementary Figure 8. The figure is too small to be read properly.

Supplementary Fig. 9 (previously Supplementary Fig. 8) has been scaled up to read properly and now covers three pages.

Reviewer #3

This paper reports a new algorithm for automated NMR resonance assignments using three NMR peak-lists. The algorithm relies (i) peak-patterns provided by a 4D TOCSY spectra for residue typing (ii) intra-residues and sequential connectivities present in 4D TOCSY and 4D NOESY spectra and (iii) expected peak positions for chemical shifts distribution. It has been applied on 4 large protein targets (from 15 to 27kDa) for which the required spectra have been recorded. The algorithm yields near complete and accurate assignments (94% of aliphatic shifts) when compared to assignments curated by a trained user on the basis of the algorithm's output. In a second stage, the generated assignment lists were combined with two 4D NOESY peak-lists for structure determination using the autoNOE-Rosetta software. As a result, generated models show a large degree of convergence. Models generated from curated or automated assignment are very similar in terms of convergence and assigned long-range NOE restraints. For the only target for which an X-ray structure is also available, it is shown that structure accuracy can be improved to 1.4Å RMSD when automatically assigned resonances from TOCSY-NOESY peak-lists are combined with restraints from two 4D NOESY peak-lists.

The whole procedure (types of spectra recorded, algorithm for automated assignment and structure calculation procedure) described in the paper is original, very efficient and potentially of great use for the NMR community since human intervention is greatly limited and the time to collect and analyze the data is drastically reduced compared to standard approaches commonly used by the community.

We thank the reviewer for her/his comments.

However, it is difficult to really assess the necessity of the approach since the results are not put in perspective. It is a method paper, so the performance must be compared to state of the art methods for automated resonance assignment and structure calculation from NOE peak-lists, e.g.:

- The FLYA algorithm (Schmidt & Guntert, 2012) is to date the most efficient tool for backbone and side-chain resonance assignment. While it is briefly mentioned in the introduction, no comparison is shown on the respective performance of FLYA and 4D CHAINS with the same input data. It can be that the algorithm presented is the only one that can do the job, but we have no idea of that.

According to the reviewer's suggestion, we performed FLYA calculations for all protein targets used in the current study and compared the assignments to those obtained using 4D-CHAINS. Here, we are performing the elaborate FLYA analysis as users and not as developers, so it is worth noting that better results might be obtained using a more optimal setup of the method. While 4D-CHAINS assignments rely exclusively on the combination of 4D-HCNH TOCSY and 4D-HCNH NOESY, the FLYA algorithm is designed to combine peak patterns from any number of input spectra. Therefore, we provided to FLYA all available spectra (4D-HCNH TOCSY, 4D-HCNH NOESY, 4D-HCCH NOESY). Based on our calculations, we found that, 4D-CHAINS outperforms FLYA consistently for all the four protein targets under study. For three proteins, namely RTT, ms6282 and Enzyme I, FLYA outputs 90% correct assignments with 7-8% error rate while for aLP target it does not produce an overall reliable assignment solution. The accuracy of FLYA for each protein target has been added to **Supplementary Fig. 8** (previously Supplementary Fig. 7) on page 10 of Supporting Information document. A detailed comparison between 4D-CHAINS and FLYA has been updated in paragraph 2 on page 4 of the revised manuscript.

- CYANA software (Guntert's group) is by far the most used approach by the community for structure calculation from NOESY spectra. There should a thorough comparison between CYANA and autoNOE-rosetta so that potential users know if it is worth resorting to the CPU-demanding Rosetta suite.

We thank the reviewer for his/her suggestion. We performed structure calculations using CYANA and have reported the results in **Supplementary Figs. 16 and 17** on pages 20 and 21 of the Supporting Information document, and also updated the text describing these results in paragraph 2 on page 7 of the revised manuscript.

Other remarks

- The title sounds a bit pompous and could be tuned down:

* “fully automated”: given the lengthy instructions given at the end of the supplementary materials on how to run CS-rossetta after 4D-CHAINS, it would be more fair to discard the “fully”

* “from 2 spectra”: it’s actually more from 4 spectra (HSQC, TOCSY, 2x NOESY) plus RDC for the 27kDa target. Maybe put instead “from a limited number of spectra”

We agree with the reviewer’s suggestion. Our title has been adapted to:

“Automated NMR resonance assignments and structure determination using a minimal set of 4D spectra”

- Details must be given on the peak-picking procedure (manual, automated then curated or fully automated ?). Also, the sensitivity to missing peaks should be discuss further.

According to the reviewer’s suggestion, we have added a section in Methods describing the full peak picking procedure we recommend to the users of our method (page 17). The sensitivity to missing peaks is illustrated in three different possible scenarios that could generate missing peaks. First, in its standard operating mode, 4D-CHAINS completes missing assignments due to the incomplete TOCSY patterns from the NOESY spectra. Second, to better evaluate the mapping performance of 4D-CHAINS, only the ^{13}C - ^1H correlated frequencies of α and β atoms were retained in the TOCSY input peaklists. Here, no mapping mistakes were made by CHAINS (**Supplementary Fig. 7**), that was able to complete the missing assignments of the remaining sidechain ^{13}C - ^1H correlations from the NOESY spectrum (**Supplementary Fig. 8**). Third, only the ^1H , ^{15}N -HSQC assignments were provided and 4D-CHAINS was asked to assign all aliphatic carbon atoms from the NOESY spectrum alone. The result of exclusively NOESY-based assignments for all four targets was 91% completeness with 5.5% error rate (**Fig. 1b**, **Supplementary Fig. 8**). The resulting *Rosetta* structures using the NOESY-based assignments are shown for aLP in **Fig. 1d**. These results are further outlined in detail in Methods, **Supplementary Table 1**, **Supplementary Figs 7 and 8**, and **Fig 1c, d**.

- Introduction : ARIA do not assign resonances (only NOE peak-lists)

We thank the reviewer for her/his suggestion. “ARIA” has been removed from the text on page 2, paragraph 1, to avoid confusion with automated resonance assignments and automated structure determination.

- RPF analysis (from Monteliones’s group) provides standardized measures to assess the reliability of an NMR model. In particular, DP-score is known to correlate very well with structure accuracy. The authors should present DP-scores for the different targets.

According to the reviewer’s suggestion, we looked into performing a complete RPF analysis to obtain DP-scores. However, we were unable to obtain DP-scores due to technical issues related to the RPF program. We therefore contacted the developers (Prof. Montelione and Yuanpeng Huang, personal communication). As per our communication, the RPF tool cannot provide reliable DP-scores using 4D data in its present form, since the DP studies were based on 3D NOESY data. Therefore, this analysis falls outside the scope of our current study and is not reported here.

- page 6: CYANA has been used with identical data as autoNOE-Rosetta for the aLP target. I don’t see any description of the CYANA setup used.

We thank the reviewer for his/her suggestion. The description of FLYA and CYANA has been updated on page 25 and 26 under the respective sub sections of the revised manuscript. Additionally, the setups used for both the methods have been updated on pages 35-36 and page 45 of the Supporting Information document.

- Conclusion: the reference to high-throughput structural genomics projects is outdated (Heineman et al. 2001).

We thank the reviewer for her/his suggestion. We updated the reference to: Vinarov, D. A. & Markley, J. L. *Expert Rev. Proteomics* 2, 49–55 (2005), in paragraph 1 on page 9 of the revised manuscript.

- Methods: “restraint violations analysis” and Supp. Table 4: A uniform upper bound of 7Å is used to measure violation of NOE restraints. What is the rationale for that ?

The choice of 7 Å as upper distance is because of 70 ms mixing time where through-space magnetization transfer between closer protons can happen within 7 Å. Additionally, we found that this holds true for confidently assigned NOEs by direct comparison with measured distances in the crystal structure of aLP (PDB ID 1P01). This point is now included in paragraph 1 on pages 27-28 of the revised manuscript.

REVIEWERS' COMMENTS:

Reviewer #1 (Remarks to the Author):

The authors have addressed my concerns satisfactorily

Reviewer #2 (Remarks to the Author):

The paper has improved significantly.
Published as it is.

Reviewer #3 (Remarks to the Author):

The authors have considerably improved the manuscript and all my previous concerns have been addressed. I thus recommend publication after considering the following final comment:

On page 2, 2nd paragraph:

“Overall, the structural ensembles calculated using autoNOE-Rosetta for all blind targets exhibit lower backbone RMSD values relative to the nearest PDB reference structures (Supplementary Fig. 17).”

When looking at Supp Fig 17, I have the impression that it might be worth mentioning in the text by how much the autoNOE-Rosetta structures are closer to the PDB reference (except for target nElt). For the 3 other targets, points seems to be quite near the diagonal (or am I misled by the size of the points).

Related to that, it could also be interesting to have the CPU times required by CYANA calculations.

Point by point response:

Reviewer #1 (Remarks to the Author):

The authors have addressed my concerns satisfactorily

We thank the reviewer for his/her valuable feedback.

Reviewer #2 (Remarks to the Author):

The paper has improved significantly.
Published as it is.

We thank the reviewer for his/her valuable suggestions that have helped us improve the manuscript.

Reviewer #3 (Remarks to the Author):

The authors have considerably improved the manuscript and all my previous concerns have been addressed. I thus recommend publication after considering the following final comment:

We thank the reviewer for his/her helpful feedback.

On page 2, 2nd paragraph:

“Overall, the structural ensembles calculated using autoNOE-Rosetta for all blind targets exhibit lower backbone RMSD values relative to the nearest PDB reference structures (Supplementary Fig. 17).”

When looking at Supp Fig 17, I have the impression that it might be worth mentioning in the text by how much the autoNOE-Rosetta structures are closer to the PDB reference (except for target nElt). For the 3 other targets, points seems to be quite near the diagonal (or am I misled by the size of the points).

Related to that, it could also be interesting to have the CPU times required by CYANA calculations.

We apologize for the confusion. The structural ensembles calculated using autoNOE-Rosetta are closer to the reference structures by (i) 0.5 Å for RTT (ii) 0.2 Å for ms6282 (iii) 0.5 Å for aLP and (iv) > 2.2 Å for nElt, compared to the structures predicted from CYANA, for both Supervised and TOCSY-NOESY assignment settings. The above results are with an exception of ms6282 protein target where CYANA structures are closer to the reference by 0.2 Å for TOCSY-NOESY assignment setting. We have highlighted this point on page 10/paragraph 2 of the revised manuscript. Additionally, the size of the points in Supp Fig 14 (earlier Supp Fig 17) corresponds to different protein targets and the area of the points is proportional to the number of residues in that protein.